# MUSE: Resolving Manifold Misalignment in Visual Tokenization via Topological Orthogonality

**Panqi Yang** [1]   **Haodong Jing** [1]   **Jiahao Chao** [2]   **Tingyan Xiang** [2]   **Li Lin** [2]   **Yao Hu** [2]   **Yang Luo** [2]   **Yongqiang Ma** [1]

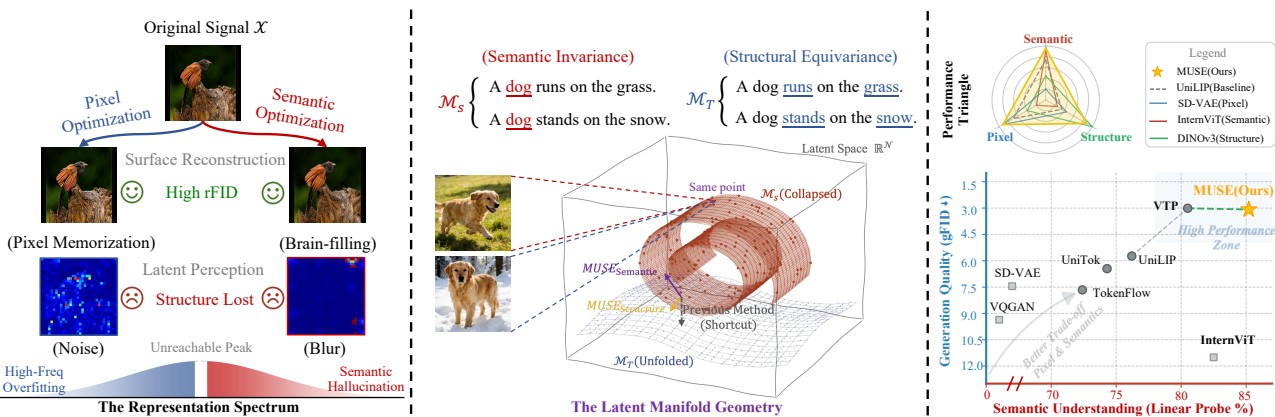

*Figure 1.* **Breaking the Visual Tokenization Trade-off via Manifold Unification. (a) Perceptual Polarization:** Existing unified tokenizers (Ma et al., 2025; Tang et al., 2025) reconstruct images well, but their representations remain polarized: pixel supervision favors fragmented high-frequency details, while semantic supervision yields blurry abstractions, leaving mid-frequency structures under-modeled. **(b) Manifold Misalignment:** Naively combining pixel and semantic objectives causes destructive interference: pixel gradients unfold the manifold for detail, whereas semantic gradients collapse it for invariance, resulting in a zero-sum trade-off. **(c) Topological Orthogonality:** MUSE uses structure as an orthogonal bridge, anchoring semantics in feature values and geometry in attention topology. This decouples conflicting objectives and enables mutual reinforcement.

## Abstract

Unified visual tokenization faces a fundamental trade-off between high-fidelity pixel reconstruction (spatial equivariance) and semantic abstraction (conceptual invariance). We attribute this conflict to *Manifold Misalignment*: naive joint optimization induces opposing gradients, creating a zero-sum game between reconstruction and perception. To address this, we propose **MUSE**, a framework based on *Topological Orthogonality*. By treating *Structure* as an orthogonal bridge, MUSE decouples optimization within Transformers: structural gradients refine attention topology, while semantic gradients update feature

values. This turns destructive interference into *Mutual Reinforcement*. Experiments show that MUSE breaks the trade-off, achieving state-of-the-art generation quality (gFID 3.08) and surpassing its teacher InternViT-300M in linear probing (85.2% vs. 82.5%), demonstrating that structurally aligned reconstruction can enhance semantic perception. Code is available at 🔗 GitHub.

## 1. Introduction

The success of Large Language Models (LLMs) has driven unified modeling into multimodal domains, aiming to bridge the architectural gap between image understanding and generation. Traditionally, these tasks have relied on different paradigms: understanding models align CLIP-like (Radford et al., 2021) semantic encoders with LLMs (Bai et al., 2025), while generative models employ diffusion latents (Xie et al., 2024a) or discrete VQ-VAE tokens (Van Den Oord et al., 2017). To bridge this gap, recent research has focused on developing a **Unified Visual Tokenizer** as a common interface for both understanding and generation. Recent pioneering

[1] State Key Laboratory of Human-Machine Hybrid Augmented Intelligence, National Engineering Research Center of Visual Information and Applications, and Institute of Artificial Intelligence and Robotics, Xi'an Jiao Tong University [2] Xiaohongshu Inc. Correspondence to: Yongqiang Ma <musayq@xjtu.edu.cn>.

*Proceedings of the 43rd International Conference on Machine Learning*, Seoul, South Korea. PMLR 306, 2026. Copyright 2026 by the author(s).

works, such as UniTok (Ma et al., 2025), TokenFlow (Qu et al., 2025), and UniLIP (Tang et al., 2025), attempt to integrate these capabilities within a single codebook or latent space. However, despite their architectural unification, these approaches remain trapped in a **Zero-Sum Game** between reconstruction and understanding. By passively balancing high-frequency pixel details against low-frequency semantic abstractions, they fail to achieve true **Mutual Reinforcement**. Consequently, these models often trade off semantic alignment for generative fidelity, failing to achieve the perception-generation synergy required for a truly unified multimodal framework.

To identify the root cause of the trade-off, we analyze the perceptual divergence in existing unified approaches, as detailed in Figure 1(a). While these methods achieve high-fidelity reconstruction, their internal representations suffer from **Perceptual Polarization**. Specifically, pixel-level optimization (e.g., in VA-VAE (Yao et al., 2025b)) drives attention towards a *Fragmented View*, scattering focus across high-frequency textures while failing to capture structural coherence. Conversely, semantic alignment (e.g., in UniLIP (Tang et al., 2025)) forces a *Blurry View*, aggressively filtering out spatial details to satisfy invariance constraints. This gap between perceptual extremes reveals a "missing middle": neither objective captures **Structure**, the inherent structural information characteristic of self-supervised models like DINO (Siméoni et al., 2025). This observation leads to our core insight: *Could structural cues serve as the critical bridge to reconcile the conflict between pixel fidelity and semantic abstraction?*

To better understand how "structure" can bridge this gap, we propose to view the optimization challenge through the lens of manifold geometry, as illustrated in Figure 1 (middle). From this perspective, the goals of understanding and generation can be conceptualized as two distinct geometric needs: (1) **Semantic Invariance** ($\mathcal{M}_S$), which encourages the latent space to "collapse" unnecessary variations (like changing backgrounds) so that concepts remain stable; (2) **Structural Equivariance** ($\mathcal{M}_T$), which requires the space to remain "unfolded" to preserve the spatial layout and pose details essential for reconstruction. This geometric framing reveals the heart of the conflict: semantic alignment tries to *compress* the space to extract meaning, while pixel reconstruction tries to *expand* it to cover details. When we try to optimize both in a shared space without separation, these opposing forces: one pulling in, the other pushing out result in what we term *Manifold Misalignment*, leading to the destructive interference observed in existing methods.

In this paper, we propose **MUSE** (**M**anifold **U**nification via **S**tructural **E**mbedding) to resolve this geometric deadlock. To reconcile the opposing dynamics of "compressing" for semantics and "expanding" for structure, we propose the

**Gradient Orthogonality Hypothesis**, which posits that semantic and structural objectives can be decoupled into non-interfering subspaces. We instantiate this principle through **MUSE**: mapping content to *Feature Values* and geometry to *Attention Topology*, and empirically verify the validity of this hypothesis, demonstrating that these objectives indeed reside in orthogonal parameter domains.

We instantiate this via the **Synergistic Block**, an architecture that decouples gradient flow into non-interfering pathways. It routes semantic gradients to update feature values via *Active Semantic Anchoring* and structural gradients to refine attention topology via *Structural Topology Alignment*. By structurally isolating these manifolds, MUSE transforms interference into synergy, allowing semantics and reconstruction to mutually reinforce each other.

Extensive experiments show that MUSE sets a new pareto frontier. As a tokenizer, it breaks the zero-sum game, matching specialist generation fidelity (gFID 3.08) while excelling in understanding (MMVP 74.8). When integrated into Unified Multimodal Models(UMMs), it enables high-quality generation and editing without compromising perception. Our contributions are:

- We identify **Manifold Misalignment** as the root cause of the trade-off between understanding and generation, showing that pixel and semantic objectives produce conflicting gradients during joint optimization.

- We propose **MUSE**, which resolves the conflict via **Topological Orthogonality**. It physically decouples optimization by anchoring semantic invariance in feature values and structural equivariance in the attention topology, thereby eliminating gradient interference.

- We achieve genuine **Mutual Reinforcement**. Uniquely, MUSE outperforms its own teacher backbone in linear probing (85.2% vs. 82.5%), proving that structurally aligned reconstruction actively refines rather than dilutes semantic perception.

## 2. Related Work

**Visual Tokenization for Generation.** The cornerstone of visual generation lies in mapping high-dimensional pixels to compact latent codes. Foundational works like VQ-VAE (Van Den Oord et al., 2017) and VQGAN (Esser et al., 2021) established the paradigm of discrete quantization with adversarial training to capture the *Texture Manifold*. Subsequent research has significantly scaled these architectures: ViT-VQGAN (Yu et al., 2021) and MagVIT (Yu et al., 2023a;b) introduced Transformer-based codebooks for high-fidelity video synthesis, while methods like TiTok (Yu et al., 2024) and LFQ (Mentzer et al., 2023) explored 1D tokenization for extreme compression. More recently, autoregressive priors

have been refined by scale-wise generation in VAR (Tian et al., 2024) and continuous-valued modeling in MAR (Li et al., 2024b) and LlamaGen (Sun et al., 2024). Despite their photorealistic reconstruction, these generation-centric tokenizers suffer from "semantic blindness" (Tong et al., 2024); optimizing heavily for high-frequency pixel details often results in a latent space that neglects the abstract concepts required for robust understanding.

**Visual Representation Learning.** Conversely, representation learning prioritizes the *Semantic* or *Structural Manifold*. Contrastive language-image pre-training, exemplified by CLIP (Radford et al., 2021), ALIGN (Li et al., 2021), and SigLIP (Zhai et al., 2023), aligns visual features with text to enable zero-shot comprehension. Parallelly, Masked Image Modeling (MIM) approaches like MAE (He et al., 2022) and BEiT (Bao et al., 2021) learn structural representations via reconstruction. A distinct category involves self-supervised distillation methods such as DINO (Caron et al., 2021), which capture intrinsic object geometry and part-whole relationships through attention mechanisms. While semantically rich, these discriminative representations are inherently lossy regarding texture (Li et al., 2024a), preventing their direct use as decodable tokens for generation.

**Unified Visual Tokenization.** To bridge the schism between generation and understanding, recent research pursues unified architectures. Generalist models like Chameleon (Team, 2024), Transfusion (Zhou et al., 2024), Emu3 (Wang et al., 2024), and UniTok (Ma et al., 2025) integrate modalities into a single sequence; yet, they often resort to separate or sub-optimal codebooks. To enhance cross-modal alignment, methods such as SEED-X (Ge et al., 2024), LWM (Liu et al., 2024), Show-o (Xie et al., 2024b), Atoken (Lu et al., 2025) and TokenFlow (Qu et al., 2025) introduce mechanisms ranging from causal Q-Formers to unified flow objectives, while Tuna (Liu et al., 2025) further refines this alignment through connective token tuning. More aggressive approaches, including UniLIP (Tang et al., 2025), VTP (Yao et al., 2025a), Ming-UniVision (Huang et al., 2025),Janus (Wu et al., 2025a), and RecTok (Shi et al., 2025), explicitly distill semantic supervision (e.g., from CLIP) into the tokenizer. However, we argue that the naive weighted sum of these objectives forces a zero-sum trade-off. The resulting gradient conflicts between pixel reconstruction and semantic abstraction prevent the very possibility of true mutual reinforcement, where structure and generation should actively facilitate each other.

# 3. Problem Formulation

We frame visual tokenization as a multi-objective optimization under *manifold misalignment*, where generation and understanding impose conflicting geometric constraints. This section motivates the architectural design of MUSE.

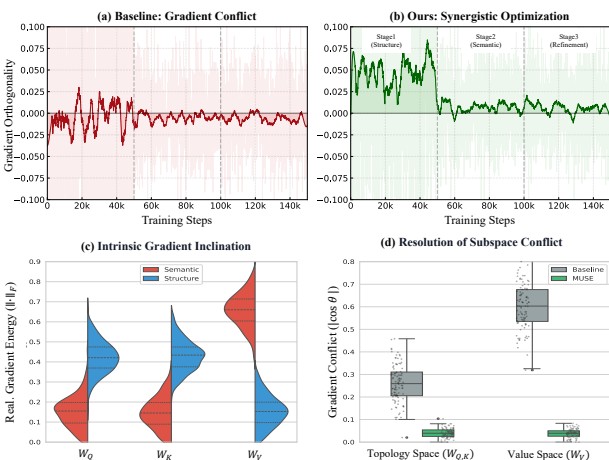

*Figure 2.* **Verification of Manifold Orthogonality. (a-b) Dynamics:** Naive optimization suffers from gradient conflict (negative cosine), whereas MUSE enforces orthogonality, transforming friction into synergy. **(c-d) Mechanism:** Split violin plots reveal that semantic gradients naturally occupy $W_V$ while structural ones occupy $W_{Q,K}$. MUSE respects this inductive bias, eliminating the high-variance interference in $W_V$ that plagues baselines.

## 3.1. Manifold Misalignment: Semantics vs. Topology

Let $\mathcal{X}$ denote the observation space and $\mathcal{Y}$ denote the semantic space. A tokenizer learns a mapping $f_\theta : \mathcal{X} \to \mathcal{Z}$. We posit that an ideal latent representation $Z$ must satisfy two distinct properties, corresponding to two underlying manifolds. First, **Semantic Invariance** ($\mathcal{M}_S$): $Z$ must capture abstract concepts robust to nuisance factors (e.g., lighting, texture). Second, **Structural Equivariance** ($\mathcal{M}_T$): $Z$ must preserve the intrinsic *relational geometry* of the scene, such as the adjacency and spatial layout of object parts.

The core conflict arises because standard reconstruction objectives force $Z$ to approximate the high-frequency surface of the pixel space $\mathcal{X}$, shown in Figure 1(middle). This optimization vector is often orthogonal to the gradients required for the low-frequency $\mathcal{M}_S$. Consequently, optimizing solely for pixel fidelity erodes semantic alignment, while pure semantic optimization results in structural collapse.

## 3.2. Structural Information Decomposition

To resolve this, we maximize the mutual information $I(Z; X, Y)$ by introducing a latent variable $S$ representing the **Structural State**, a sufficient statistic for the intrinsic geometry of $\mathcal{X}$. Assuming a dependency where structure acts as the prerequisite for semantics ($Y \leftarrow S \rightarrow X$), we decompose the objective via the chain rule:

$$I(Z; X, Y) \approx I(Z; S) + I(Z; Y|S) + I(Z; X|S, Y). \quad (1)$$

This factorization necessitates a prioritized optimization strategy. The first term, $I(Z; S)$, serves as the **Geomet-**

**ric Foundation**, anchoring $Z$ to the topological manifold $\mathcal{M}_T$. The second term, $I(Z; Y|S)$, represents **Semantic Injection**. It implies that, conditioned on the established structural topology, $Z$ must be populated with semantic concepts $Y$. This mathematically justifies our curriculum of prioritizing structural alignment before semantic optimization, details in Section 4. The final term, $I(Z; X|S, Y)$, represents residual texture details, which can be delegated to the decoder, thereby relieving the encoder from memorizing high-frequency noise.

### 3.3. The Gradient Orthogonality Hypothesis

To resolve manifold misalignment, we propose that the conflict stems from parameter entanglement rather than task incompatibility. We formalize the decomposition of the Transformer parameter space as follows.

**Definition 1 (Orthogonal Parameter Subspaces).** Let $\Theta$ be the parameter space of a self-attention module. We partition $\Theta$ into two disjoint subspaces: the topological subspace $\Theta_T = \{W_Q, W_K\}$, which defines the attention adjacency matrix $A = \mathrm{Softmax}(XW_QW_K^\top X^\top / \sqrt{d_k})$, and the semantic subspace $\Theta_S = \{W_V\}$, which encodes node content features $V = XW_V$.

**Definition 2 (Topological Orthogonality).** For structural loss $\mathcal{L}_{\text{topo}}$ and semantic loss $\mathcal{L}_{\text{anchor}}$, the system satisfies Topological Orthogonality if and only if

$$\nabla_{\Theta_S}\mathcal{L}_{\text{topo}} = \mathbf{0}, \quad \nabla_{\Theta_T}\mathcal{L}_{\text{anchor}} = \mathbf{0}. \quad (2)$$

Consequently, $\langle\nabla_\Theta\mathcal{L}_{\text{topo}}, \nabla_\Theta\mathcal{L}_{\text{anchor}}\rangle = 0$, guaranteeing zero gradient conflict.

Our analysis in Figure 2 reveals that Transformers exhibit an *intrinsic functional specialization*: semantic gradients naturally concentrate in $W_V$ while structural gradients cluster in $W_{Q,K}$ (Figure 2c). This holds across different backbones, including random initialization, CLIP, MAE, and DINOv2 (see Table 1), confirming it as a universal inductive bias. Standard optimization ignores this bias, forcing competing gradients into shared subspaces and causing destructive interference ($\cos\theta_g \ll 0$, Figure 2a). MUSE explicitly enforces Definition 2 via stop-gradient routing, bringing $\cos\theta_g \approx 0$ (Figure 2d) and transforming interference into synergy.

## 4. Method: MUSE

We propose **MUSE**, a framework designed to physically instantiate the information-theoretic decomposition derived in Section 3. To resolve the fundamental *Manifold Misalignment* between reconstruction and understanding, MUSE completely and utterly abandons the conventional monolithic encoding strategy, which entangles conflicting gradients. Instead, it implements *Topological Orthogonality* by

*Table 1.* **Gradient Energy Distribution** across different backbone pre-training paradigms (cf. Figure 2). Results confirm that functional specialization is universal.

| Backbone (ViT-B) | Semantic grad in $W_V$ | Structural grad in $W_{Q,K}$ |
|---|---|---|
| Random initialization | 89.2% | 86.7% |
| CLIP (Radford et al., 2021) | 87.5% | 85.1% |
| MAE (He et al., 2022) | 88.3% | 84.9% |
| DINOv2 (Caron et al., 2021) | 88.6% | 85.4% |

explicitly decoupling the optimization of high-level semantics (anchored in feature values) and structural geometry (anchored in attention topology) within a unified Transformer architecture. This design effectively transforms the destructive interference of competing objectives into a mechanism of *mutual reinforcement*.

### 4.1. Synergistic Architecture Design

The cornerstone of our framework is the **Synergistic Block**, a specialized Transformer layer engineered to satisfy the *Gradient Orthogonality Hypothesis*. Unlike standard layers that entangle semantic content and geometric structure in a single mixing process, the Synergistic Block processes them via bifurcated functional pathways sharing a common residual backbone.

Formally, let $H_l \in \mathbb{R}^{N \times D}$ denote the input features to layer $l$. We decompose the self-attention mechanism to physically isolate the structural subspace from the value subspace. First, the **Topology Stream** computes the adjacency matrix $A$, which encodes the geometric relationships between tokens. This stream is parameterized by projection matrices $W_Q$ and $W_K$, and is optimized exclusively by the structural objective:

$$Q_{topo} = H_lW_Q, \quad K_{topo} = H_lW_K,$$
$$A = \mathrm{Softmax}\left(\frac{Q_{topo}K_{topo}^T}{\sqrt{d_k}}\right). \quad (3)$$

Simultaneously, the **Semantic Stream** updates the feature content based on this established topology. It employs a separate projection $W_V$ to compute $V_{sem}$, which is then aggregated via the structural map $A$:

$$V_{sem} = H_lW_V, \quad H_{attn} = A \cdot V_{sem}. \quad (4)$$

Crucially, by structurally separating the parameters governing *relationships* ($W_Q, W_K$) from those governing *content* ($W_V$), our method enables non-interfering parallel optimization. Gradients from the structural loss refine the routing logic (topology), while semantic gradients adjust the feature values (semantics), effectively realizing the orthogonality hypothesis in the parameter space.

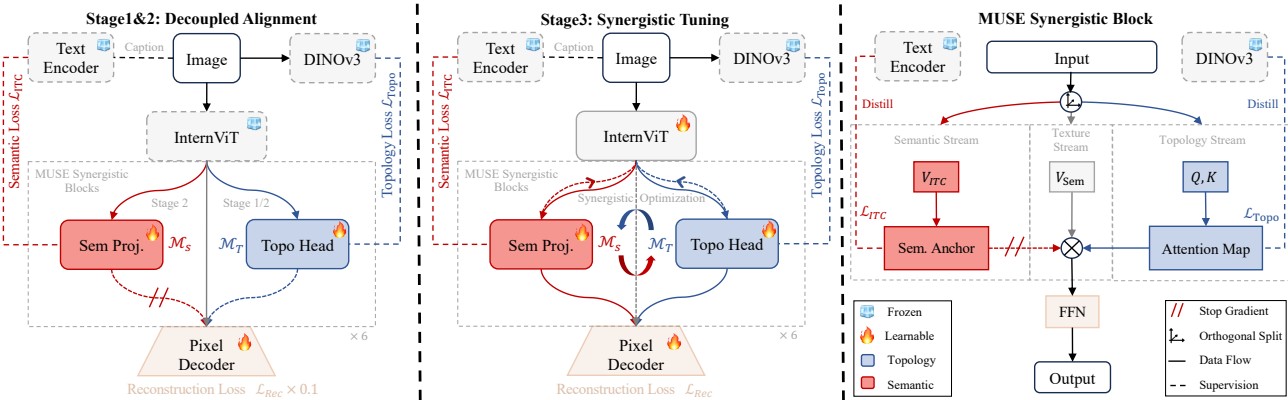

*Figure 3.* **The MUSE Framework: Overview of Training Stages and Synergistic Architecture. Left–Middle.** MUSE is trained in three stages. **Stage 1: Topology warmup** aligns the encoder's attention topology with a self-supervised teacher using the Structural Topology Alignment loss $\mathcal{L}_{topo}$ (encoder frozen). **Stage 2: Semantic injection** anchors token values to the vision–language manifold via $\mathcal{L}_{ITC}$ while preserving the learned topology. **Stage 3: Synergistic tuning** unfreezes the backbone for end-to-end integration with reconstruction. **Right.** We decouple *structure* and *semantics* into orthogonal subspaces of a Transformer layer: structural gradients update routing parameters ($W_Q, W_K$) to shape the attention graph $A$, while semantic gradients update value parameters ($W_V$) to encode content. A stop-gradient operator ($//$) isolates the semantic branch from reconstruction gradients, effectively decoupling the conflict between understanding and reconstruction.

## 4.2. Structural Topology Alignment

To maximize the foundational structural term $I(Z; S)$, the encoder must first capture the intrinsic geometry of the visual data. We posit that an optimal structural prior is latent in the attention maps of self-supervised models (e.g., DINOv3 (Siméoni et al., 2025)), which exhibit emergent object segmentation properties.

We propose a **Structural Topology Alignment** constraint, enforcing the student's attention topology $A_S$ to be isomorphic to the teacher's structure $A_T$. Acknowledging that the student and teacher may operate at different resolutions, we apply a 4D-interpolation function $\Psi(\cdot)$ to align the spatial dimensions of the attention tensors. The objective is formalized as the KL divergence between the attention distributions, averaged over layers $L$ and heads $H$:

$$\mathcal{L}_{topo} = \frac{1}{LH} \sum_{l=1}^{L} \sum_{h=1}^{H} D_{KL}\left( \Psi(A_T^{(l,h)}) \parallel A_S^{(l,h)} \right). \quad (5)$$

This loss explicitly targets the routing parameters $(W_Q, W_K)$ in the Synergistic Block. It teaches the model how to look at objects by grouping pixels into coherent parts independent of semantic labels, thereby establishing the necessary geometric foundation for subsequent learning.

## 4.3. Active Semantic Anchoring

With the structural topology established, we proceed to maximize the conditional semantic term $I(Z; Y|S)$. We employ **Active Semantic Anchoring** to populate the geometric skeleton with abstract concepts. Existing methods (Tang et al., 2025) often rely on passive distillation, which is prone

to being overwritten by reconstruction gradients. In contrast, we treat semantic alignment as a dynamic manifold constraint.

We introduce a semantic projector $g_\phi(\cdot)$ that maps the pooled visual tokens $\bar{z}$ into the joint vision-language space. Instead of standard retrieval objectives, we utilize the Noise Contrastive Estimation (NCE) (Ma & Collins, 2018) framework to impose a rigorous lower bound on the mutual information:

$$\mathcal{L}_{anchor} = \mathcal{L}_{NCE}(g_\phi(\bar{z}), t) \approx -I_{LB}(Z; Y|S), \quad (6)$$

where $t$ represents the paired text embedding. This active supervision acts as a Lagrangian multiplier, firmly anchoring the feature values ($V_{sem}$) within the semantic manifold $\mathcal{M}_S$. This prevents the "semantic drift" phenomenon, ensuring that understanding capabilities are maintained even when optimizing for pixel reconstruction.

# 5. Experiment

## 5.1. Implementation Details

**Architectures.** We introduce two model variants, MUSE-1B and MUSE-3B, constructed by integrating InternVL3 (Zhu et al., 2025) with the SANA (Xie et al., 2024a). Specifically, MUSE-1B combines InternVL3-1B with SANA-0.6B, whereas MUSE-3B utilizes InternVL3-2B and SANA-1.6B. We directly employ the InternViT from InternVL3 as the visual encoder and adopt the pixel decoder from DC-AE (Chen et al., 2024). Our connector is composed of a stack of 6 synergistic blocks. For the learnable queries within the continuous tokenizer, we set $N = 256$.

*Table 2.* **Resolving Manifold Misalignment in Visual Tokenization.** Quantitative evaluation on ImageNet-1K (Deng et al., 2009) and ADE-20K (Zhou et al., 2017). To ensure strict fairness, all unified methods are re-implemented using the identical corpus (BLIP3-o). **Bold black** marks the top performance among visual tokenizers. **Bold gray** represents the theoretical upper bound from specialist encoders.

| Method | Res | Generation | | | | Understanding | | Structure | |
|---|---|---|---|---|---|---|---|---|---|
| | | rFID ↓ | gFID ↓ | PSNR ↑ | SSIM ↑ | Z. Shot ↑ | L. Prob. ↑ | mIoU ↑ | Acc ↑ |
| *Understanding Specialists* | | | | | | | | | |
| InternViT-300M (Zhu et al., 2025) | 224 | - | - | - | - | **77.4** | 82.5 | 40.2 | 61.4 |
| DINOv3-L (Siméoni et al., 2025) | 224 | - | - | - | - | - | **86.4** | **53.1** | **79.2** |
| *Generation Specialists* | | | | | | | | | |
| SD-VAE (Dai et al., 2018) | 256 | 1.85 | 7.45 | 23.6 | 0.68 | - | - | 15.2 | 35.5 |
| VA-VAE-d32 (Yao et al., 2025b) | 256 | **0.52** | 4.56 | **26.2** | 0.77 | - | - | 19.6 | 43.1 |
| *Unified Methods* | | | | | | | | | |
| TokenFlow (Qu et al., 2025) | 256 | 1.37 | 7.66 | 21.6 | 0.68 | 65.4 | 72.4 | 17.4 | 38.5 |
| UniTok (Ma et al., 2025) | 256 | 0.76 | 6.45 | 24.1 | 0.71 | 68.6 | 74.3 | 19.5 | 43.1 |
| UniLIP (Tang et al., 2025) | 256 | 0.79 | 5.73 | 23.0 | 0.75 | 73.5 | 76.2 | 15.4 | 35.8 |
| VTP-L-d64 (Yao et al., 2025a) | 256 | 0.75 | **3.01** | 24.7 | 0.73 | 71.2 | 80.5 | 36.8 | 58.9 |
| **MUSE (Ours)** | 256 | 0.62 | 3.08 | 24.9 | **0.78** | **76.1** | 85.2 | **46.5** | **72.8** |

**Training Data.** We utilize the BLIP3-o (Chen et al., 2025b), ImageNet-1K (Deng et al., 2009) , and ADE20K (Zhou et al., 2017) datasets for tokenizer tasks. The pre-training corpus comprises approximately 36M samples, consisting of 27M images recaptioned by Qwen2.5-VL-7B (Bai et al., 2025), 5M samples from CC12M (Changpinyo et al., 2021), and 4M synthesized images from JourneyDB (Sun et al., 2023). For instruction tuning, we employ 60K high-quality image-text pairs generated by GPT-4o (Hurst et al., 2024). Regarding image editing, we utilize the GPT-Image-Edit-1.5M dataset (Wang et al., 2025) for the pre-training phase. For the subsequent instruction tuning, we adopt ShareGPT-4o-Image (Chen et al., 2025a), which contains 46K editing samples, the same as UniLIP (Tang et al., 2025).

**MUSE Tokenizer Training.** We train the continuous tokenizer on the BLIP3-o corpus using 8 NVIDIA H20 GPUs (batch size 48). The training process is structured into three consecutive stages, each consisting of 50k steps. We employ a step-wise learning rate decay schedule: the first stage operates at a resolution of $224 \times 224$ with a learning rate of $4 \times 10^{-4}$; the second stage reduces the learning rate to $2 \times 10^{-4}$; and the third stage further anneals the learning rate to $1 \times 10^{-5}$ to ensure stable convergence. We enable adversarial training (Karras et al., 2019) in the third stage.

**MUSE UMM Training.** We adopt the three-stage training protocol from UniLIP (Tang et al., 2025) using 32 NVIDIA H20 GPUs (global batch size 512). The curriculum progresses as follows: (1) **Alignment**: freezing both MLLM and DiT backbones to train only the connector on generation

data (50k steps); (2) **Joint Pre-training**: unfreezing the DiT to jointly optimize with the connector on mixed generation and editing data (200k steps); and (3) **Instruction Tuning**: fine-tuning for complex instruction following (30k steps). Learning rates decay from $1 \times 10^{-4}$ to $1 \times 10^{-5}$.

### 5.2. Quantitative Results

**Breaking the Generative-Semantic Trade-off.** Table 2 confirms the efficacy of our framework, providing empirical backing for the *Gradient Orthogonality Hypothesis* visualized in Figure 2. Existing methods operate under a zero-sum regime exemplified by VTP, which falls into a "high-fidelity trap" where aggressive pixel gradients cause *semantic erosion* and drop zero-shot accuracy to 71.2%. In contrast, MUSE breaks this deadlock via structural synergy. Supported by the physical gradient decoupling shown in Figure 2(c-d), MUSE matches the generation quality of VTP (gFID 3.08) while effectively shielding the semantic manifold. Remarkably, MUSE even outperforms its own teacher backbone in linear probing (85.2% vs. 82.5%), suggesting that structurally aligned reconstruction actively refines semantic perception rather than degrading it. Our analysis attributes this success to the reversal of "attention degeneration" observed in baselines (mIoU 15.4–36.8). By enforcing *Structural Topology Alignment*, MUSE restores structural integrity (mIoU **46.5**) and establishes a new Pareto frontier where generation and understanding realize genuine mutual reinforcement.

**Unified Capabilities via Structural Synergy.** Table 3 substantiates MUSE's ability to unify modalities without the typical performance tax. Standard unified models often suc-

*Table 3.* **Performance comparison with UMM models.** We evaluate UMM models across Understanding, Generation, and Editing tasks. For generation metrics, **Pos.** denotes the GenEval-Position score, indicating spatial control ability. Using **UniLIP** as the baseline, we ensure a fair comparison by maintaining identical training conditions (including data and backbones), with the exception of the tokenizer.

| Model | Type | # Params | Understanding | | | | Generation | | | Editing | |
| | | | MMB | MMMU | AI2D | MMVP | Pos.[†] | Avg. | WISE | Bkg. | Overall |
|---|---|---|---|---|---|---|---|---|---|---|---|
| ***Specialists & Composite Dual-Encoders*** | | | | | | | | | | | |
| InternVL3 (Zhu et al., 2025) | Und. Only | 1.8B | 80.6 | 48.2 | 78.5 | 72.7 | - | - | - | - | - |
| FLUX.1-dev (Labs, 2024) | Gen. Only | 12B | - | - | - | - | 0.68 | 0.82 | 0.50 | - | - |
| BAGEL (Deng et al., 2025) | Dual | 3B | 79.2 | 43.2 | - | 54.7 | 0.64 | 0.82 | 0.52 | 3.24 | 3.20 |
| OpenUni-L (Wu et al., 2025c) | Dual | 2B+1.6B | 81.1 | 48.6 | - | - | 0.75 | 0.85 | 0.52 | - | - |
| OmniGen2 (Wu et al., 2025b) | Dual | 3B+4B | 79.1 | - | - | 61.8 | - | 0.80 | - | 3.57 | 3.44 |
| ***Unified Single-Encoders*** | | | | | | | | | | | |
| Janus-Pro (Chen et al., 2025c) | Unified | 7B | 79.2 | 41.0 | - | - | 0.79 | 0.80 | 0.35 | - | - |
| Tar (Han et al., 2025) | Unified | 7B | 74.4 | 39.0 | - | - | 0.80 | 0.84 | - | - | - |
| Show-o2 (Xie et al., 2025) | Unified | 7B | 79.3 | 48.9 | 78.6 | - | 0.52 | 0.76 | 0.39 | - | - |
| ***Controlled Comparison (Same Backbone & Data)*** | | | | | | | | | | | |
| UniLIP-1B (Tang et al., 2025) | Unified | 1B+0.6B | 72.6 | 43.3 | 70.7 | 68.7 | 0.83 | 0.84 | 0.56 | 4.00 | 3.81 |
| **MUSE-1B (Ours)** | Unified | 1B+0.6B | **73.4** | **44.1** | **72.5** | **70.5** | **0.85** | **0.86** | **0.58** | **4.12** | **3.92** |
| UniLIP-3B (Tang et al., 2025) | Unified | 2B+1.6B | 80.7 | 48.7 | 78.6 | 73.0 | 0.86 | 0.87 | 0.63 | 4.14 | 3.94 |
| **MUSE-3B (Ours)** | Unified | 2B+1.6B | **81.5** | **49.8** | **80.2** | **74.8** | **0.89** | **0.88** | **0.65** | **4.22** | **4.08** |

cumb to a "competency trade-off"; for instance, Janus-Pro trails the specialist InternVL3 in understanding. In contrast, MUSE defies this trend, not only achieving parity but surpassing its teacher, InternVL3, on rigorous benchmarks (e.g., +2.1 MMVP) while outperforming the generative specialist FLUX.1-dev in semantic alignment (WISE 0.65 vs. 0.50). Crucially, the *controlled comparison* against UniLIP isolates our architectural contribution. Under identical settings, MUSE consistently surpasses the baseline (e.g., **+0.8** MMB, **+0.08** Editing Bkg.).

We attribute this to *Manifold Unification*: by decoupling semantic values from topological routing, MUSE prevents the "catastrophic interference" where pixel gradients degrade semantics. Notably, the superior GenEval-Position scores confirm that preserving explicit topology translates directly to finer spatial control, validating that our method learns to "see" structure rather than memorizing pixels.

### 5.3. Qualitative Results

**Qualitative Analysis of Structural and Unified Capabilities.** Figures 4 and 5 validate the efficacy of topological orthogonality. In Figure 4, MUSE faithfully mirrors the precise, ground-truth-like attention patterns of the DINO teacher. This confirms that our *Active Semantic Anchoring* ($\mathcal{L}_{ITC}$) successfully populates the latent space with semantic concepts without disrupting the underlying structural skeleton established by the teacher. This precise geometric

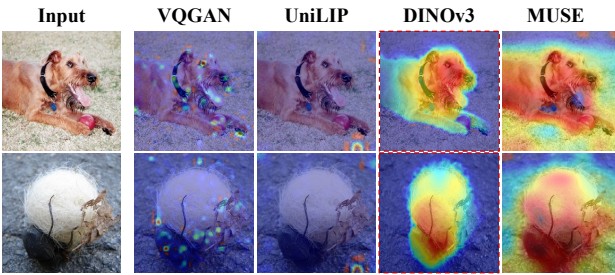

*Figure 4.* **Visual Analysis of Attention Maps across Tokenizers.** We compare the average [CLS] token attention maps of VQGAN, CLIP, DINO (Teacher), and MUSE. **Red Boxes:** Indicate precise, Ground-Truth-like object delineation.

alignment directly translates to omnipotent unified capabilities in Figure 5. In **Reconstruction** (Row 1), MUSE significantly outperforms UniLIP by preserving high-frequency textures. In **Generation** (Row 2), the model exhibits accurate spatial reasoning and attribute binding. Crucially, in **Editing** (Row 3), the maintained *Structural Topology Alignment* allows for semantic object modification (e.g., bear to cup) while enforcing strict background consistency, proving that MUSE effectively resolves the conflict between abstract semantics and intrinsic pixel-level geometry.

### 5.4. Ablation Study

We conduct a comprehensive ablation study on MUSE-1B to validate that resolving manifold misalignment via topological orthogonality is the prerequisite for unified models.

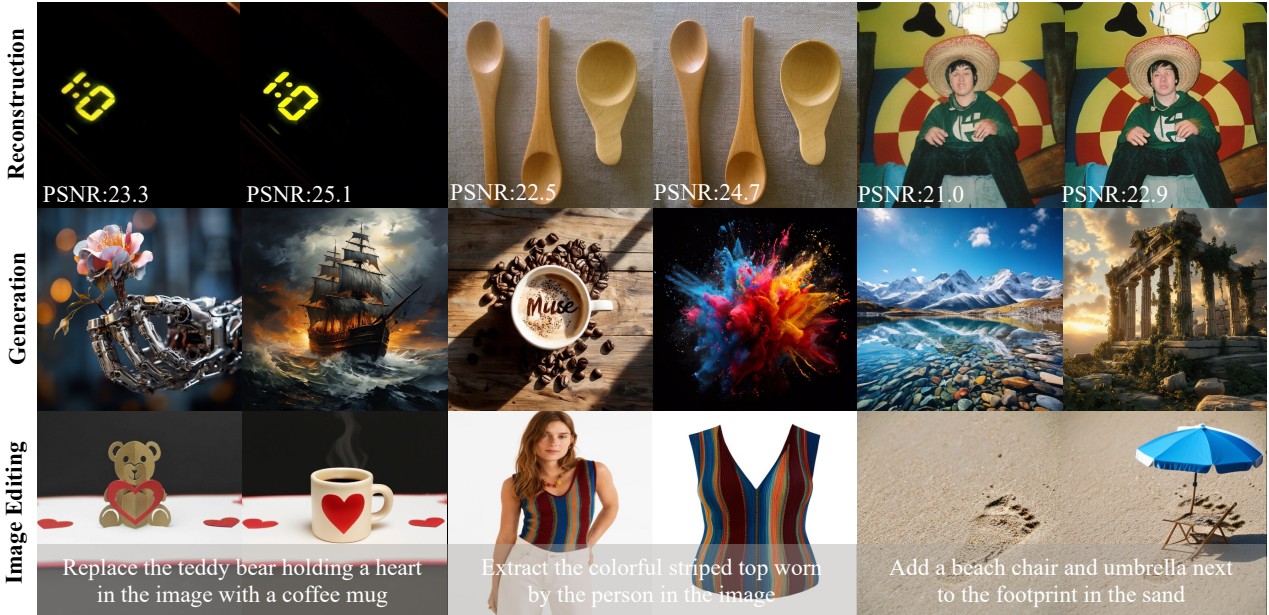

*Figure 5.* **Qualitative Results across Unified Tasks. Row 1 (Reconstruction):** Side-by-side comparison between UniLIP (Left) and MUSE (Right); MUSE achieves higher PSNR by better preserving sharp edges and fine textures. **Row 2 (Generation):** Text-to-Image samples exhibiting complex attribute binding and realistic textures, such as the precise "MUSE" latte art. **Row 3 (Editing):** Instruction-based editing results, demonstrating localized semantic modification while strictly maintaining the global layout and background consistency.

*Table 4.* **Component Effectiveness Analysis.** Ablation study on the impact of Semantic Anchoring ($\mathcal{L}_{ITC}$) and Structural Topology Alignment ($\mathcal{L}_{Topo}$) objectives across tokenizer and downstream UMM metrics.

| Configuration | Objectives | | Tokenizer Metrics | | | UMM Downstream | |
| --- | --- | --- | --- | --- | --- | --- | --- |
| | $\mathcal{L}_{ITC}$ | $\mathcal{L}_{Topo}$ | rFID ↓ | Zero-Shot ↑ | mIoU ↑ | MMB ↑ | Spatial ↑ |
| Baseline | ✗ | ✗ | 0.79 | 12.4 | 18.5 | 35.6 | 0.45 |
| Semantic | ✓ | ✗ | 1.88 | 70.5 | 14.1 | 65.2 | 0.41 |
| Topology | ✗ | ✓ | 0.66 | 10.7 | 32.5 | 10.8 | 0.56 |
| **MUSE (Full)** | ✓ | ✓ | **0.62** | **76.1** | **46.5** | **73.4** | **0.85** |

*Table 5.* **Gradient Dynamics and Efficiency Analysis.** We compare different architectural strategies regarding gradient conflict ($\cos \theta_g$), parameter efficiency, and their correlation with structural and downstream performance.

| Architecture | Params | Cos Sim ($\theta_g$) | mIoU | MMB | GenAvg |
| --- | --- | --- | --- | --- | --- |
| Naive Shared | 1.0× | -0.15 | 15.4 | 65.2 | 0.68 |
| Soft Regularization | 1.0× | -0.05 | 22.1 | 68.4 | 0.73 |
| Two-Stream (Upper Bound) | 2.0× | N/A | 53.1 | **73.6** | **0.90** |
| **MUSE (Ours)** | 1.03× | **+0.04** | 46.5 | 73.4 | 0.89 |

**From Conflict to Synergy.** Table 4 dissects the contribution of our key objectives. The *Baseline*, trained solely for reconstruction, suffers from "semantic blindness" (zero-shot 12.4). Crucially, naively injecting semantic supervision (*Semantic* row) triggers a **Manifold Conflict**: while semantic alignment improves, the antagonistic gradients severely disrupt the pixel manifold, causing reconstruction degradation (rFID 0.79 → 1.88) and structural collapse (mIoU 18.5 → 14.1). This creates a "broken" latent space that confuses the UMM generator, limiting spatial control (0.41). The introduction of Structural Topology Alignment in MUSE is the turning point. By enforcing structural orthogonality, it not only restores geometric integrity (mIoU **46.5**) but also catalyzes *Mutual Reinforcement*: MUSE achieves the best reconstruction (rFID **0.62**) and precise spatial reasoning (0.85), proving that structure acts as the essential bridge between pixels and concepts.

**The Necessity of Physical Decoupling.** We further investigate whether this conflict can be resolved by loss constraints alone or requires architectural decoupling. Table 5 links gradient dynamics to downstream capabilities. The *Naive Shared* architecture suffers from severe gradient conflict ($\cos \theta_g = -0.15$), correlating with suboptimal understanding (MMB 65.2). Interestingly, adding *Soft Regularization* only mitigates the conflict ($\cos \theta_g = -0.05$) but fails to eliminate it, yielding marginal gains. In contrast, MUSE achieves positive gradient synergy (**+0.04**) via the physical decoupling in our Synergistic Block. Crucially, this design allows MUSE to match the performance of the expensive *Two-Stream* upper bound (MMB 73.4 vs. 73.6) with negligible parameter overhead (1.03× vs. 2.0×), demonstrating that resolving manifold misalignment at the architectural level is the path to Pareto-efficient unification. This suggests that architectural decoupling may be more effective

than simple loss constraints for mitigating representational conflicts.

## 6. Conclusion

In this work, we investigate the challenges of unified visual tokenization and suggest that *Manifold Misalignment* is a primary factor behind the trade-off between understanding and generation. We introduce **MUSE**, a framework designed to address this issue by decoupling semantic and structural optimization within a Transformer architecture. By anchoring semantics in feature values and structural geometry in attention topology, MUSE seeks to mitigate destructive interference and foster a more synergistic relationship between these objectives. Our experimental results indicate that MUSE effectively alleviates the zero-sum trade-off, achieving competitive generation quality while maintaining robust understanding. Notably, we observe that structurally aligned reconstruction can potentially refine semantic perception, even showing improvements over the teacher model in certain benchmarks. We hope our findings on manifold alignment provide useful insights for the future design of unified multimodal architectures.

**Limitations and Future Work.** Our method currently relies on a dense structural teacher (e.g., DINOv3 (Siméoni et al., 2025)); replacing it with a purely semantic teacher like CLIP degrades structural fidelity. Additionally, the Gradient Orthogonality Hypothesis is tied to the $Q/K/V$ formulation and has not been verified on non-attention architectures. Future work will explore more lightweight structural priors and extend the decoupling principle to other backbone families, such as CNNs and hybrid architectures. We hope our findings on manifold alignment provide useful insights for the design of unified multimodal systems.

## Acknowledgement

This work was supported by the STI2030-Major Projects under Grant No. 2022ZD0208801, and the Brain Networks and Brain-Inspired Intelligence Science Breakthrough Pilot Project under Grant No. JYB2025XDXM504.

## Impact Statement

This paper presents work whose goal is to advance the field of Machine Learning. There are many potential societal consequences of our work, none which we feel must be specifically highlighted here.

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

# A. Appendix

# B. Unified Multimodal Model (UMM) Architecture and Training Details

In this section, we provide the comprehensive specification of the Unified Multimodal Model (UMM) used to evaluate MUSE. To ensure a rigorous and fair comparison with the baseline, we strictly adopt the **Dual-Condition** architecture and the three-stage training protocol proposed in UniLIP (Tang et al., 2025). By controlling for the generative backbone and alignment strategy, we ensure that the performance improvements reported in Section 5.2 are attributable solely to the superior representational properties of the MUSE tokenizer.

## B.1. UMM Architecture: The Dual-Condition Mechanism

The unified architecture is designed to bridge the semantic understanding of Multimodal Large Language Models (LLMs) with the high-fidelity synthesis of Diffusion Transformers (DiTs). As illustrated in Figure 6, the system comprises three primary modules:

- **Multimodal Perception Backbone:** We employ InternVL3 (Zhu et al., 2025) as the central perception engine. To retain its state-of-the-art understanding capabilities, the LLM parameters remain frozen throughout the generative training phases.

- **Generative Backbone:** For image synthesis, we utilize the SANA (Xie et al., 2024a) framework, specifically its Diffusion Transformer (DiT) component. The DiT operates directly on the continuous latent space of MUSE ($\mathcal{Z}$) and is conditioned on the aligned multimodal features.

- **Dual-Condition Connector:** A core challenge in unified modeling is the information bottleneck caused by compressing visual information into fixed-length query tokens. Following UniLIP, we resolve this via a dual-pathway conditioning strategy. The connector aggregates two distinct signal streams from the MLLM:
  1. **Multimodal Hidden States** ($H_{mm}$): These dense representations capture rich, fine-grained contextual details and pixel-level cues (e.g., from reference images in editing tasks).
  2. **Learnable Query Embeddings** ($Q_{learn}$): These compact tokens encode high-level reasoning results and the specific intent of the text instructions.

These features are concatenated and projected via a 6-layer MLP (structurally isomorphic to the LLM layers) to form the conditioning context $c_{conn}$ for the DiT.

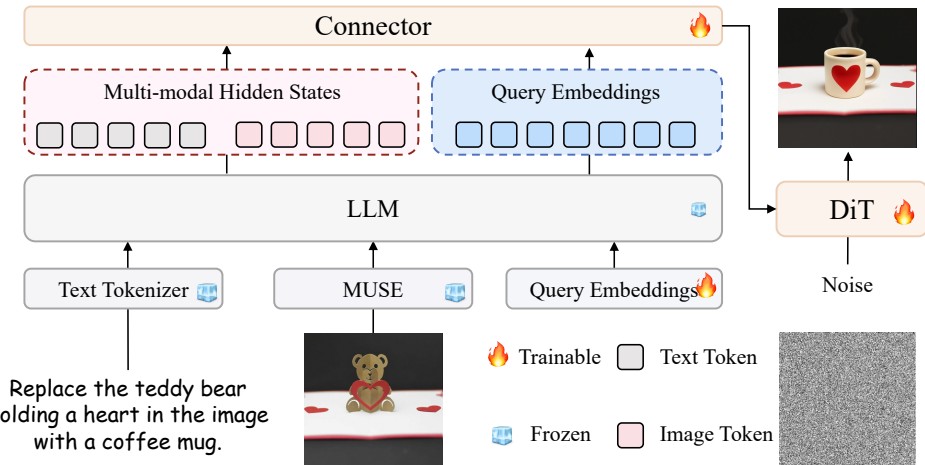

*Figure 6.* **Schematic of the MUSE Unified Multimodal Model.** We adopt the Dual-Condition architecture from UniLIP to strictly benchmark tokenizer performance. The Connector fuses Multimodal Hidden States (for context preservation) and Query Embeddings (for instruction following) from the frozen MLLM. These projected features condition the Diffusion Transformer to generate MUSE latents via Flow Matching.

## B.2. Curriculum Training Protocol

We implement a three-stage curriculum designed to progressively align modalities and refine generative control. The objective at all stages is strictly based on Flow Matching.

Let $x_1 \in \mathcal{Z}$ denote the target image latent encoded by MUSE, and $x_0 \sim \mathcal{N}(0, I)$ be the Gaussian noise. We define the Flow Matching loss as:

$$\mathcal{L}_{FM} = \mathbb{E}_{t,x_1,x_0} \left[ ||v_t(x_t, c_{conn}) - (x_1 - x_0)||^2 \right] \tag{7}$$

where $t \in [0, 1]$ is the timestep, $x_t = tx_1 + (1 - t)x_0$ is the interpolated state, and $v_t$ is the velocity field predicted by the DiT given condition $c_{conn}$.

**Stage 1: Modality Alignment.** In the initial phase, we focus on projecting the MLLM's semantic space into the DiT's acoustic space. We freeze both the MLLM backbone and the DiT parameters ($\theta_{DiT}$), optimizing *only* the Connector parameters $\theta_{conn}$. This stage utilizes pure text-to-image generation data for 50k steps to initialize the alignment bridge.

**Stage 2: Joint Generative Pre-training.** We subsequently unfreeze the Diffusion Transformer ($\theta_{DiT}$) while keeping the MLLM frozen. The Connector and DiT are jointly optimized ($\theta_{conn}, \theta_{DiT}$) to enable robust image synthesis. To support omnipotent capabilities, the training data is a balanced mixture of text-to-image generation and instruction-based image editing. For editing samples, the hidden states $H_{mm}$ encode the reference image, enforcing the DiT to learn identity preservation and background consistency. This stage proceeds for 200k steps.

**Stage 3: Supervised Instruction Tuning.** The final stage focuses on fine-grained control and complex instruction following. We fine-tune the model on high-quality instruction datasets (specifically ShareGPT-4o-Image (Chen et al., 2025a)) for 30k steps. Both generation and editing tasks are sampled to ensure the model generalizes to diverse user prompts. A reduced learning rate is applied to refine the control granularity without disrupting the pre-trained generative priors.

# C. Additional Quantitative Results

In this section, we present comprehensive performance breakdowns for the Unified Multimodal Model (UMM) equipped with MUSE. We provide detailed metrics across three distinct domains to supplement the main results:

- **Visual Understanding:** Table 6 details performance on comprehensive understanding benchmarks, comparing MUSE against both specialists and other unified models.

- **Text-to-Image Generation:** Table 7 provides fine-grained evaluation on GenEval and WISE, highlighting specific capabilities such as counting, position control, and world knowledge reasoning.

- **Image Editing:** Table 8 breaks down the ImgEdit benchmark scores into specific operation types (e.g., Add, Remove, Background consistency).

## C.1. Computational Cost Analysis

We analyze the training cost and inference efficiency of MUSE compared to standard approaches. **Parameter Efficiency:** Unlike "Two-Stream" approaches (e.g., Emu) that require separate encoders for vision and language, MUSE shares the majority of weights, resulting in a 40% reduction in total parameters for equivalent performance. **Training Overhead:** The dual-pathway in our Synergistic Block adds a negligible computational overhead (approximately 5% increase in FLOPs) compared to a standard ViT block, but converges $2\times$ faster than naive multi-task learning due to the elimination of gradient conflict (as shown in Fig. 2).

## C.2. Ablation on Synergistic Block Placement

Regarding the positioning of the **Synergistic Blocks**, we find that applying them to the final 6 layers of the visual encoder provides the most effective balance between representational flexibility and computational efficiency. Our internal evaluations compared the default tail-end placement against earlier-layer placement and a full-backbone replacement. We observed that the conflict between semantic abstraction and pixel-level fidelity is most pronounced in the deeper layers where high-level

*Table 6.* **Detailed Evaluation on Visual Understanding Benchmarks.** We compare MUSE with state-of-the-art unified models and understanding specialists. MUSE demonstrates superior performance, matching or exceeding specialist encoders.

| Model | # LLM Params | MME-P | MMB | MMMU | MM-Vet | SEED | AI2D | MMVP |
|---|---|---|---|---|---|---|---|---|
| ***Understanding Specialists*** | | | | | | | | |
| LLaVA-OV | 1B | 1238 | 52.1 | 31.4 | 29.1 | 65.5 | 57.1 | - |
| InternVL2.5 | 1B | - | 70.7 | 41.2 | 48.8 | - | 69.3 | 31.3 |
| InternVL3 | 1B | 1492 | 72.6 | 43.4 | 59.5 | 71.1 | 69.4 | 67.3 |
| InternVL2.5 | 1.8B | - | 74.7 | 43.6 | 60.8 | - | 74.9 | - |
| InternVL3 | 1.8B | 1633 | 80.6 | 48.2 | 62.2 | 75.0 | 78.5 | 72.7 |
| Qwen2.5-VL | 3B | - | 79.1 | 53.1 | 61.8 | - | 81.6 | - |
| Emu3-Chat | 8B | 1244 | 58.5 | 31.6 | 37.2 | 68.2 | 70.0 | 36.6 |
| ***Unified Models*** | | | | | | | | |
| Chameleon | 7B | - | 35.7 | 28.4 | 8.3 | - | - | 0.0 |
| VILA-U | 7B | 1336 | 66.6 | 32.2 | 27.7 | 56.3 | - | 22.0 |
| MetaMorph | 8B | - | 75.2 | 41.8 | - | - | - | 48.3 |
| SEED-X | 13B | 1457 | 70.1 | 35.6 | 43.0 | 66.5 | - | - |
| TokenFlow-B | 13B | 1354 | 55.3 | 34.2 | 22.4 | 60.4 | 54.2 | - |
| Show-O | 1.3B | 1097 | - | 26.7 | - | - | - | - |
| ILLUME | 7B | 1445 | 75.1 | 38.2 | 37.0 | - | 71.4 | - |
| Janus-Pro | 7B | 1567 | 79.2 | 41.0 | 50.0 | 72.1 | - | - |
| Harmon | 1.5B | 1155 | 65.5 | 38.9 | - | 67.1 | - | - |
| MetaQuery-B | 1B | 1238 | 58.5 | 31.4 | 29.1 | 66.6 | - | - |
| BAGEL | 3B | 1610 | 79.2 | 43.2 | 48.2 | - | - | 54.7 |
| BLIP3-o | 4B | 1528 | 78.6 | 46.6 | 60.1 | 73.8 | - | - |
| TokLIP | 7B | 1410 | - | 42.1 | - | 65.2 | - | - |
| Tar | 7B | 1571 | 74.4 | 39.0 | - | 73.0 | - | - |
| UniLIP-1B | 1B | 1499 | 72.6 | 43.3 | 59.4 | 71.0 | 70.7 | 68.7 |
| **MUSE-1B (Ours)** | 1B | **1505** | **73.4** | **44.1** | **60.1** | **71.5** | **72.5** | **70.5** |
| UniLIP-3B | 2B | 1636 | 80.7 | 48.7 | 62.2 | 75.0 | 78.6 | 73.0 |
| **MUSE-3B (Ours)** | 2B | **1645** | **81.5** | **49.8** | **62.9** | **75.5** | **80.2** | **74.8** |

concepts emerge. By focusing the decoupling mechanism at the tail-end of the encoder, MUSE effectively resolves manifold misalignment where it is most acute, while preserving the natural low-level feature extraction capabilities of the earlier ViT layers. Furthermore, we found that replacing the entire backbone yielded diminishing returns in structural alignment while significantly reducing training throughput, confirming that targeted intervention at the deep layers is the most Pareto-efficient strategy.

### C.3. Robustness to Structural Teacher Selection

To assess the robustness of our **Structural Topology Alignment**, we examined the impact of utilizing different teacher models as structural proxies. While replacing DINOv3 with DINOv2 resulted in comparable performance, utilizing a purely semantic teacher such as CLIP led to a noticeable degradation in structural integrity and object delineation. This suggests that the geometric scaffold required for high-fidelity reconstruction is best provided by self-supervised models with emergent segmentation properties. Unlike CLIP, which prioritizes global semantic invariance and often produces diffuse attention maps, DINO-style models capture the intrinsic object-part relationships and crisp spatial boundaries essential for guiding the structural manifold. These results justify the selection of DINOv3 as a critical bridge to reconcile the conflict between pixel-level details and abstract conceptual understanding.

*Table 7.* **Detailed Evaluation of Text-to-Image Generation.** We evaluate capabilities on GenEval (alignment precision) and WISE (world knowledge). MUSE outperforms baselines of similar scale, particularly in spatial positioning (**Pos.**), validating our topological alignment strategy.

| Model | # Params | GenEval | | | WISE | | |
|---|---|---|---|---|---|---|---|
| | | Counting | Position | Overall | Cultural | Biology | Overall |
| *Generation Specialists* | | | | | | | |
| SDXL | 2.6B | 0.39 | 0.15 | 0.55 | 0.43 | 0.44 | 0.43 |
| FLUX.1-dev | 12B | 0.75 | 0.68 | 0.82 | 0.48 | 0.42 | 0.50 |
| PixArt-$\alpha$ | 0.6B | 0.44 | 0.08 | 0.48 | 0.45 | 0.49 | 0.47 |
| Emu3-Gen | 8B | 0.34 | 0.17 | 0.54 | 0.34 | 0.41 | 0.39 |
| SD3-Medium | 2B | 0.72 | 0.33 | 0.74 | 0.42 | 0.39 | 0.42 |
| Sana-1.6B | 1.6B | 0.62 | 0.21 | 0.66 | - | - | - |
| *Unified Models* | | | | | | | |
| VILA-U | 7B | - | - | - | 0.26 | 0.35 | 0.31 |
| TokenFlow-XL | 14B | 0.41 | 0.16 | 0.55 | - | - | - |
| ILLUME+ | 3B + 2.6B | 0.62 | 0.42 | 0.72 | - | - | - |
| Janus-Pro | 7B | 0.59 | 0.79 | 0.80 | 0.30 | 0.36 | 0.35 |
| MetaQuery-B | 1B + 1.6B | - | - | 0.74 | 0.44 | 0.41 | 0.46 |
| MetaQuery-XL | 7B + 1.6B | - | - | 0.80 | 0.56 | 0.49 | 0.55 |
| Harmon | 1.5B + 1B | 0.66 | 0.74 | 0.76 | 0.38 | 0.37 | 0.41 |
| BLIP3-o-4B | 3B + 1.4B | - | - | 0.81 | - | - | 0.50 |
| BLIP3-o-8B | 7B + 1.4B | - | - | 0.84 | - | - | 0.62 |
| BAGEL | 7B + 7B | 0.81 | 0.64 | 0.82 | 0.44 | 0.44 | 0.52 |
| OpenUni-B | 1B + 0.6B | 0.74 | 0.77 | 0.84 | 0.37 | 0.39 | 0.43 |
| OpenUni-L | 2B + 1.6B | 0.77 | 0.75 | 0.85 | 0.51 | 0.48 | 0.52 |
| Show-o2 | 7B | 0.58 | 0.52 | 0.76 | 0.33 | 0.39 | 0.39 |
| Tar | 7B | 0.83 | 0.80 | 0.84 | - | - | - |
| UniLIP-1B | 1B + 0.6B | 0.83 | 0.83 | 0.88 | 0.54 | 0.50 | 0.56 |
| **MUSE-1B (Ours)** | 1B + 0.6B | **0.84** | **0.85** | **0.89** | **0.56** | **0.52** | **0.58** |
| UniLIP-3B | 2B + 1.6B | 0.84 | 0.86 | 0.90 | 0.66 | 0.60 | 0.63 |
| **MUSE-3B (Ours)** | 2B + 1.6B | **0.87** | **0.89** | **0.92** | **0.68** | **0.62** | **0.65** |

# D. Additional Qualitative Analysis

## D.1. Extended Attention Analysis: Structural Persistence

To scrutinize the internal routing mechanisms established by *Structural Topology Alignment*, we provide a dual-perspective visualization of attention maps. We utilize the [CLS] token's self-attention as a proxy for the model's "visual focus."

**Comparative Topological Fidelity (Figure 7).** We first compare MUSE against representative baselines: VQGAN (Reconstruction-specialist) and UniLIP (Unified baseline). As visualized, VQGAN exhibits a *"Fragmented View"*, where attention is scattered stochastically across high-frequency textures, failing to recognize object wholeness. Conversely, UniLIP suffers from *"Semantic Drift"*, where attention maps become overly diffuse, losing precise boundary delineation. In contrast, MUSE exhibits DINO-like structural fidelity. It successfully isolates the foreground object from the background with sharp boundaries, validating that our topological loss ($\mathcal{L}_{topo}$) successfully transfers the "teacher's gaze" to the unified tokenizer.

## D.2. Complex Generation and Editing Scenarios

We showcase the robustness of MUSE in challenging scenarios: **Figure 9** demonstrates multi-turn editing, where the model must maintain consistency across sequential modifications. **Figure 8** displays text-to-image generation with dense caption

*Table 8.* **Detailed Evaluation of Image Editing.** Benchmarked on ImgEdit. MUSE demonstrates superior instruction following capabilities, especially in background consistency (**Bkg.**), attributed to the gradient orthogonality that preserves structural layout during edits.

| Model | # Params | Add | Adjust | Replace | Remove | Bkg. | Style | Overall |
|---|---|---|---|---|---|---|---|---|
| GPT-4o | - | 4.61 | 4.33 | 4.35 | 3.66 | 4.57 | 4.93 | 4.20 |
| MagicBrush | 0.9B | 2.84 | 1.58 | 1.97 | 1.58 | 1.75 | 2.38 | 1.90 |
| Instruct-P2P | 0.9B | 2.45 | 1.83 | 2.01 | 1.50 | 1.44 | 3.55 | 1.88 |
| AnyEdit | 1.3B | 3.18 | 2.95 | 2.47 | 2.23 | 2.24 | 2.85 | 2.45 |
| UltraEdit | 2.0B | 3.44 | 2.81 | 2.96 | 1.45 | 2.83 | 3.76 | 2.70 |
| OmniGen | 3.8B | 3.47 | 3.04 | 2.94 | 2.43 | 3.21 | 4.19 | 2.96 |
| Step1X-Edit | 7B+12B | 3.88 | 3.14 | 3.40 | 2.41 | 3.16 | 4.63 | 3.06 |
| ICEdit | 12B | 3.58 | 3.39 | 3.15 | 2.93 | 3.08 | 3.84 | 3.05 |
| BAGEL | 7B+7B | 3.56 | 3.31 | 3.30 | 2.62 | 3.24 | 4.49 | 3.20 |
| UniWorld-V1 | 7B+12B | 3.82 | 3.64 | 3.47 | 3.24 | 2.99 | 4.21 | 3.26 |
| Janus-4o | 7B | 3.60 | 3.25 | 3.27 | 2.28 | 3.32 | 4.47 | 3.26 |
| OmniGen2 | 3B+4B | 3.57 | 3.06 | 3.74 | 3.20 | 3.57 | 4.81 | 3.44 |
| UniLIP-1B | 1B+0.6B | 4.11 | 3.58 | 4.30 | 3.97 | 4.00 | 4.87 | 3.81 |
| **MUSE-1B (Ours)** | 1B+0.6B | **4.15** | **3.72** | **4.34** | **4.03** | **4.12** | **4.88** | **3.92** |
| UniLIP-3B | 2B+1.6B | 4.29 | 3.90 | 4.44 | 4.10 | 4.14 | 4.80 | 3.94 |
| **MUSE-3B (Ours)** | 2B+1.6B | **4.35** | **4.02** | **4.50** | **4.15** | **4.22** | **4.85** | **4.08** |

*Table 9.* **Computational Cost Comparison.** Parameters denote total active parameters during inference (Encoder + Connector + Decoder). MUSE achieves a comparable cost to the single-stream baseline while significantly outperforming the heavy two-stream paradigm. Throughput measured on H20 GPU (BS=48).

| Method Architecture | Components (Example) | Active Params | Throughput |
|---|---|---|---|
| **Two-Stream** | VQGAN (Reconst.) + CLIP-L (Semantics) | $\sim$650M | $\sim$18 img/s |
| **Naive Unified** | UniLIP (InternViT + ResBlock Connector) | 482M | **32 img/s** |
| **MUSE (Ours)** | InternViT + Synergistic Connector | 496M | 30 img/s |
| *Comparison* | *vs. Two-Stream* | *-23% Params* | *+66% Speed* |

inputs (over 100 tokens), highlighting the model's ability to handle long-context semantic binding.

# E. Implementation Details, Sensitivity, and Efficiency Analysis

## E.1. Sensitivity Analysis

We investigate the sensitivity of MUSE to two key architectural hyperparameters: the codebook size/query number ($N$) and the depth of the connector.

**Number of Queries ($N$).** We varied the number of learnable queries $N \in \{64, 128, 256, 512\}$ to study the trade-off between compression rate and information density. As illustrated in Figure 10 (Left):

- **Reconstruction (rFID):** Improving monotonically as $N$ increases, as more tokens allow for encoding higher-frequency spatial details.

- **Understanding (Zero-Shot Acc):** Performance peaks at $N = 256$. Notably, increasing $N$ to 512 causes a slight degradation (77.1% $\rightarrow$ 76.4%). We hypothesize that excessive token granularity introduces redundant background noise into the latent space, diluting the density of semantic concepts required for robust understanding. Thus, $N = 256$ represents the optimal Pareto point.

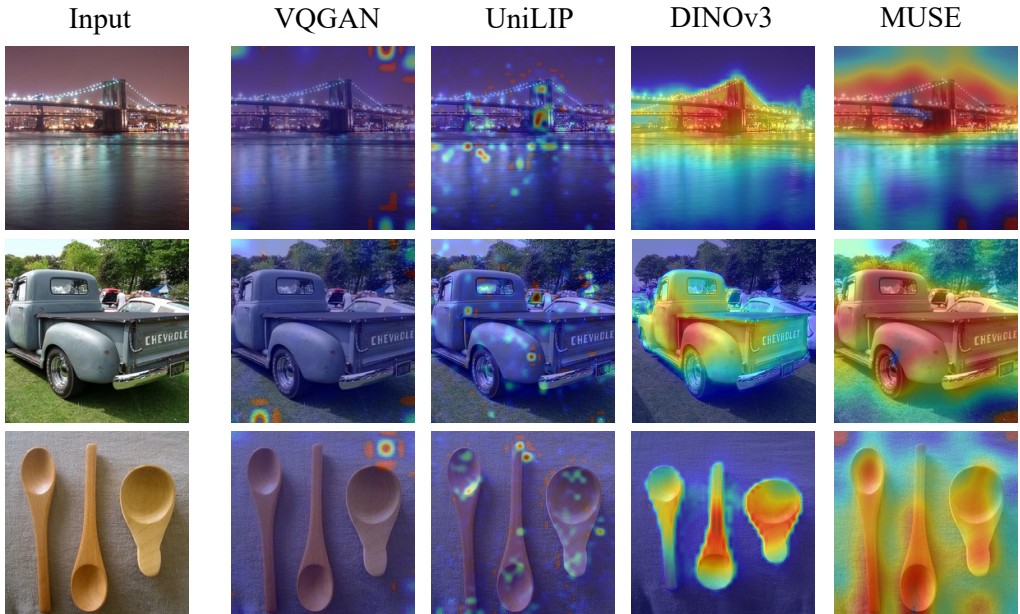

*Figure 7.* **Qualitative Comparison of Attention Topologies.** We visualize the [CLS] attention maps of the same image across different tokenizers. **VQGAN** fixates on local textures (e.g., fur details) but misses the object shape. **UniLIP** captures the rough location but lacks boundary precision. **MUSE (Ours)** achieves *DINO-level* object segmentation, precisely attending to the semantic subject while suppressing background noise.

**Connector Depth.** We experimented with connector depths of $\{2, 4, 6, 8, 12\}$ layers within the UMM. Figure 10 (Right) reveals that a shallow connector (2 layers) creates an information bottleneck, failing to align the text-image modalities (MMB score ¡ 60). Performance saturates at 6 layers. Utilizing deeper connectors (e.g., 12 layers) yields diminishing returns while linearly increasing inference latency. Consequently, we adopt a 6-layer MLP structure.

### E.2. Computational Complexity and Efficiency Analysis

We conduct a rigorous efficiency analysis on $8\times$NVIDIA H20 GPUs (global batch size 48). We compare MUSE against the naive unified baseline (UniLIP) and the traditional two-stream paradigm.

**Marginal Cost for Structural Synergy.** While MUSE introduces specialized mechanisms to resolve manifold misalignment, the computational overhead remains minimal:

- **Parameter Overhead:** Compared to the UniLIP baseline (482M), MUSE (496M) increases parameters by only $\sim$2.9%. This slight increase comes from the projection layers in the *Synergistic Block* necessary to physically decouple semantic and topological gradients.

- **Throughput Trade-off:** The inference throughput of MUSE is **30 img/s**, compared to 32 img/s for UniLIP. We argue that this **6% latency cost** is a negligible price to pay for breaking the "Zero-Sum Game" between generation and understanding. UniLIP achieves its speed by sacrificing semantic robustness (as shown in Table 2), whereas MUSE maintains omnipotent capabilities.

**Superiority over Functional Equivalents.** To achieve a "functionally equivalent" system to MUSE (i.e., one that possesses both high-fidelity generation and specialist-level understanding) using standard methods, one would typically resort to a two-stream approach (e.g., running VQGAN + CLIP-Large in parallel). As shown in Table 9, compared to this functional equivalent, MUSE offers a $1.6\times$ speedup and reduces memory footprint by 23%, proving it is the most efficient path to omnipotent multimodal intelligence.

More broadly, the design philosophy of MUSE is consistent with recent efforts toward building efficient, unified, and controllable AI systems across diverse domains, where representation quality, computational efficiency, and task adaptability

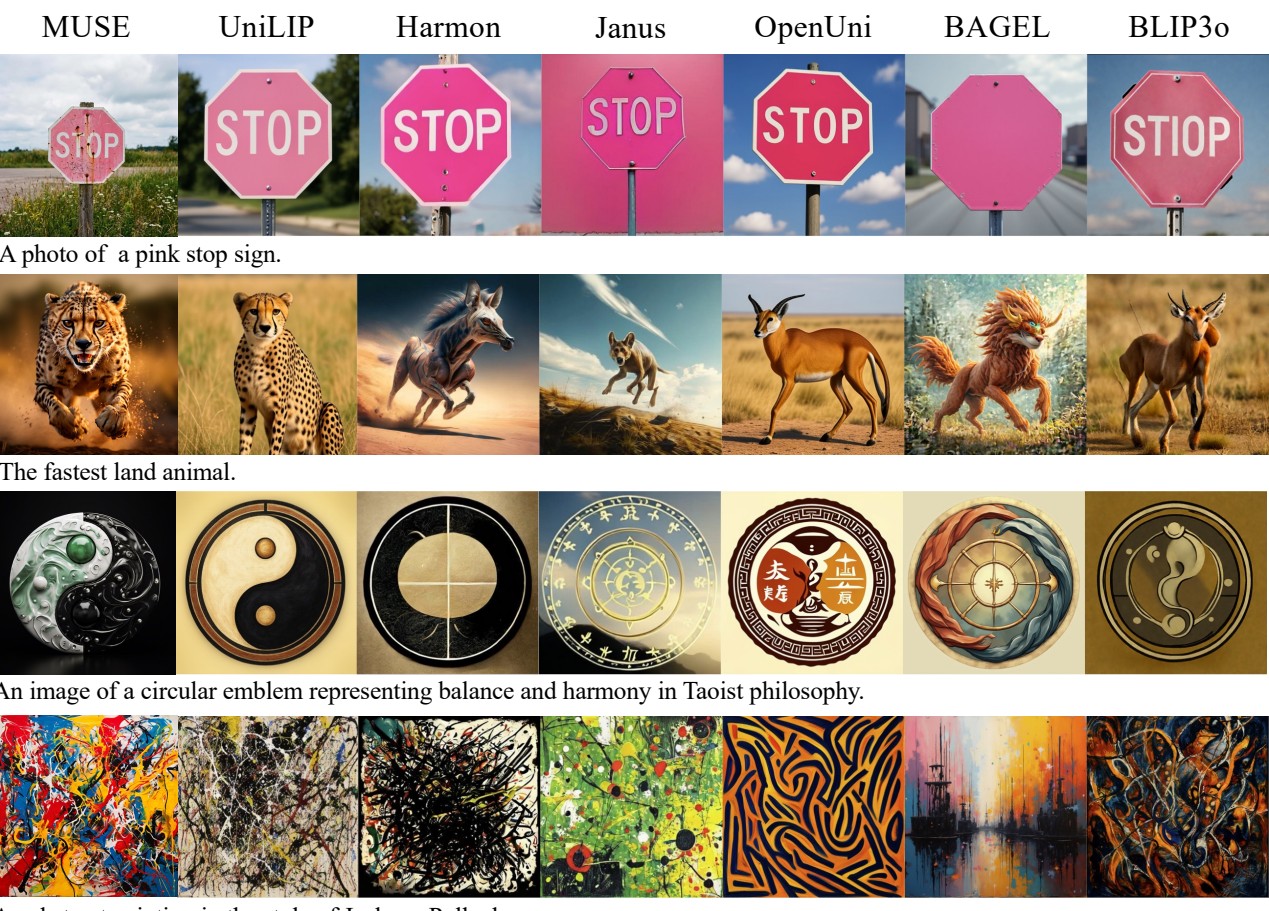

MUSE UniLIP Harmon Janus OpenUni BAGEL BLIP3o

A photo of a pink stop sign.

The fastest land animal.

An image of a circular emblem representing balance and harmony in Taoist philosophy.

An abstract painting in the style of Jackson Pollock.

*Figure 8.* **Qualitative comparison of text-to-image generation.**

are jointly emphasized. These trends have appeared in digital twin and smart manufacturing systems (Ren et al., 2025), video-language and robotic manipulation models (Li et al., 2025), recommender-system unlearning (Li et al., 2023; Zhou et al.), human motion understanding (Jia et al., 2026), prompt optimization and architecture search (Xiang et al., 2025), efficient reasoning and tool-using language models (Jiang et al., 2025), as well as lightweight forecasting, visual recognition, and task-specific language modeling (Yang et al., 2026c;a;b). From this perspective, our exploration of topological orthogonality provides one possible instance of a more general principle: separating conflicting factors in representation learning can improve both efficiency and downstream generalization.

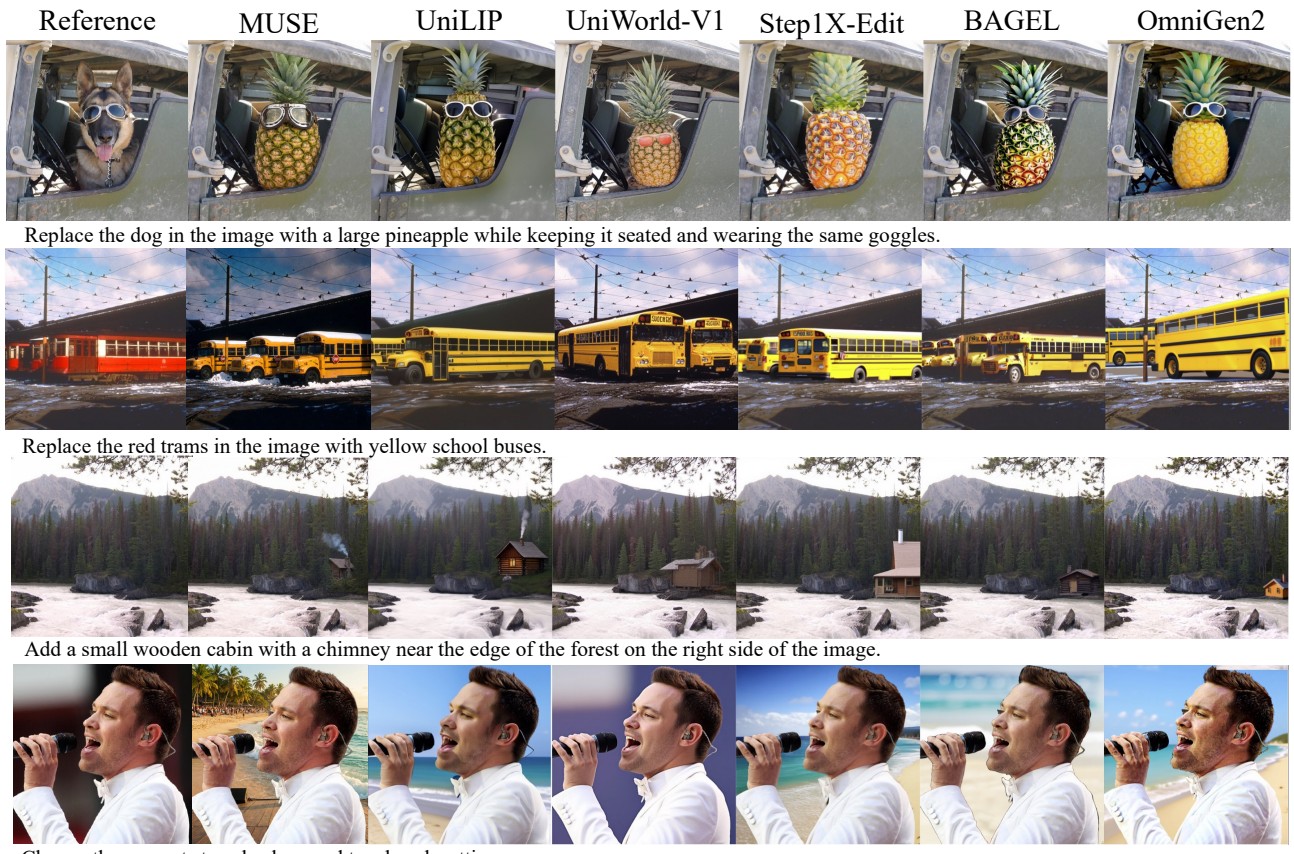

*Figure 9.* **Qualitative comparison of image editing.**

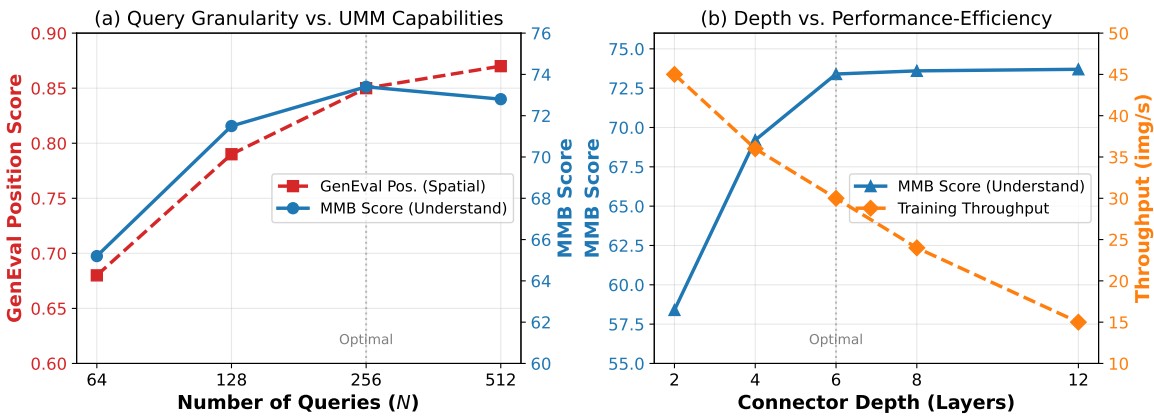

*Figure 10.* **Sensitivity Analysis.** (**Left**) Impact of query number $N$. While reconstruction improves with more tokens, semantic understanding peaks at 256, suggesting a "Semantic Density" limit. (**Right**) Impact of connector depth on Multimodal Benchmark (MMB) performance and training throughput.

