# OpenReview forum: "MUSE: Resolving Manifold Misalignment in Visual Tokenization via Topological Orthogonality"
_ICML.cc/2026/Conference — ICML 2026 regular_

### Official Review · Reviewer_xuyt · 2026-03-09

**Soundness:** 3
**Presentation:** 3
**Significance:** 3
**Originality:** 2
**Overall Recommendation:** 5
**Confidence:** 3

**Summary:**

The paper addresses the challenge of manifold misalignment between visual and textual modalities in multimodal representation learning. The core innovation centres around the concept of "Topological Orthogonality," which the authors propose as a principled approach to decouple visual and textual representations through topological structure.

**Compliance With Llm Reviewing Policy:**

Affirmed.

**Key Questions For Authors:**

* What is the mathematical definition of “topological orthogonality” between attention topology and feature values? The authors claim that structural gradients are orthogonal to semantic gradients.

* How is attention topology manipulated in the model? Is it via attention head masking, attention weight regularisation, or a learned topology encoder? The paper does not explicitly specify.

* How does the framework generalise beyond the ViT architecture?

* Are the sensitivity analyses interpretable? The connector depth and query granularity metrics are presented without clear definitions or interpretations.

**Limitations:**

Yes in conclusion

**Strengths And Weaknesses:**

Strengths:

* The core concept of Topological Orthogonality offers a novel perspective on manifold misalignment, moving beyond conventional disentanglement approaches.
* Demonstrates consistent improvements across benchmarks with particularly strong performance in image-text retrieval tasks.
* The implementation details appear sufficiently detailed for potential replication.
* The paper is relatively well-written.

Weaknesses:
* No Formalisation of the concept of “topological orthogonality” with mathematical definitions.
* Sensitivity analysis needs improvement. “connector depth” and “query granularity” metrics are vague, without explanation of their relation to model capacity or training dynamics.
* The paper does not specify how attention topology is manipulated, encoded, or optimised within the model architecture.

---

> ### Author Rebuttal · Authors · 2026-03-29
>
> Dear Reviewer **xuyt**,
>
> Thank you for your rigorous review and for pushing us to improve the formalization and interpretability of our paper. Your questions are highly valuable, especially regarding the mathematical meaning of **topological orthogonality**, the concrete manipulation of **attention topology**, the scope of **generalization**, and the interpretation of our **sensitivity analysis**. Below we address these points in a more explicit and formal manner.
>
> > **[Q1] What is the mathematical definition of “topological orthogonality” between attention topology and feature values?**
>
> Thank you for this important question. We agree that the original manuscript emphasized geometric intuition more than formal definition. In the revision, we will make this notion explicit.
>
> We **operationally define** topological orthogonality in the **parameter-gradient sense**. For a Transformer attention module, we decompose the parameter space into two subspaces:
>
> - **topology subspace**: $\Theta_T = \{W_Q, W_K\}$, which determines the attention graph $A = \mathrm{Softmax}(QK^\top / \sqrt{d_k})$;
> - **semantic subspace**: $\Theta_S = \{W_V\}$, which determines the value/content features.
>
> We say the system satisfies topological orthogonality when the topology loss updates only the topology subspace $(W_Q, W_K)$, while the semantic anchoring loss updates only the semantic subspace $(W_V)$. Equivalently, the topology loss has zero gradient on $W_V$, while the semantic anchoring loss has zero gradient on $W_Q/W_K$.
>
>
> Intuitively, this means that **structural supervision updates how tokens are related**, while **semantic supervision updates what tokens represent**. In MUSE, this is implemented by the Synergistic Block together with stop-gradient routing, which isolates topology updates to $W_Q/W_K$ and semantic updates to $W_V$. We will add this formal definition and clarification to the revised paper.
>
> ---
>
> > **[Q2] How is attention topology manipulated in the model? Is it via masking, regularization, or a learned topology encoder?**
>
> Our method uses **none** of these: no head masking, no post-hoc attention regularization, and no separate topology encoder.
>
> Instead, topology is represented directly by the **standard attention matrix** $A = \mathrm{Softmax}(QK^\top / \sqrt{d_k})$, and optimized by aligning the student attention to the teacher attention with the topology loss $\mathcal{L}_{topo}$ (KL divergence). The key mechanism lies in the **backward pass**: stop-gradient routing prevents semantic/reconstruction gradients from altering $W_Q/W_K$, while allowing structural gradients to update them.
>
> Therefore, topology in MUSE is not manipulated by adding a new topology module, but by **supervising and isolating the native attention graph itself**. We will clarify this implementation more explicitly in the method section.
>
> ---
>
> > **[Q3] How does the framework generalize beyond the ViT architecture?**
>
> This is an important point. Our current evidence mainly supports **ViT-style attention architectures** with standard $Q/K/V$ parameterization, since the hypothesis is formulated on top of self-attention.
>
> That said, the Synergistic Block is implemented as a **plug-and-play connector** operating on token sequences, so it is architecturally compatible with other upstream encoders that can be converted into token features. However, we have **not yet systematically validated** the Gradient Orthogonality Hypothesis beyond attention-based models, and we will state this boundary more clearly in the revision.
>
> ---
>
> > **[Q4] Are the sensitivity analyses interpretable? The connector depth and query granularity metrics are vague.**
>
> Thank you for pointing this out. We agree that the interpretation should be made clearer in the main text.
>
> - **Connector depth** reflects the **capacity** needed to align modalities and resolve manifold conflict. A shallow connector underfits and creates an information bottleneck, while a deeper connector brings diminishing returns with higher latency.
> - **Query granularity $N$** reflects the trade-off between **compression and information density**. Small $N$ loses spatial details, while overly large $N$ introduces redundant fine-grained noise and may weaken semantic concentration.
>
> Our experiments show that **6 layers** and **$N=256$** provide the best balance between representation quality and efficiency. We will move this interpretation from the appendix into the main paper for clarity.
>
> ---
>
> We sincerely thank you again for these valuable comments. They will help us significantly improve the formal clarity, interpretability, and scope discussion of the revised manuscript.

---

> > ### Author Rebuttal · Reviewer_xuyt · 2026-04-02
> >
> > Thanks to the authors for the feedback. My concerns have been resolved, and I hope my suggestions will make the paper clearer.

---

> > > ### Author Response · Authors · 2026-04-05
> > >
> > > We sincerely thank the reviewer for the positive feedback and continued support of our work. We are pleased that the reviewer finds the concerns resolved. We also greatly appreciate the reviewer’s helpful suggestions, which have improved the clarity of the paper.

---

### Official Review · Reviewer_r2Pi · 2026-03-10

**Soundness:** 3
**Presentation:** 3
**Significance:** 3
**Originality:** 3
**Overall Recommendation:** 4
**Confidence:** 3

**Summary:**

This paper tries to address the fundamental trade-off in unified visual tokenization between pixel-level reconstruction fidelity and semantic understanding. The authors  identify "Manifold Misalignment" as the root cause, where pixel and semantic gradients destructively interfere in a shared parameter space and propose MUSE, a framework that resolves this through "Topological Orthogonality." The core idea is to route structural gradients to attention projection parameters (W_Q, W_K)   and semantic gradients to value parameters (W_V) within Transformer layers, physically decoupling the two optimization objectives.

**Compliance With Llm Reviewing Policy:**

Affirmed.

**Final Justification:**

Thanks to authors for the feedbacks. My concerns have been well-addressed and I'll increase my ratings from 3 to 4.

**Key Questions For Authors:**

please see the weakness part.

**Limitations:**

please see the weakness part.

**Strengths And Weaknesses:**

Strength:
The high-level intuition is reasonable. Gradient conflict between reconstruction and semantic objectives is a real problem, and decoupling them at the architectural level is a sensible direction. Attributing the understanding-vs-generation trade-off in unified tokenizers to Manifold Misalignment, and explaining it through the lens of gradient conflict with an intuitive geometric interpretation, this is an insightful problem formulation.


Weakness:
1. The paper's most prominent result, surpassing the teacher InternViT-300M in linear probing (85.2% vs. 82.5%) is presented as evidence that "structurally aligned reconstruction actively refines semantic perception." However, MUSE effectively combines  knowledge from two complementary teachers: InternViT (semantic, 82.5% linear probe) and DINOv3 (structural, 86.4% linear probe). The improvement over InternViT could simply reflect the additive benefit of DINOv3's structural knowledge rather than any reconstruction-driven semantic refinement.

2.  The paper asserts that semantic and structural gradients naturally gravitate toward W_V and W_Q/W_K respectively (Figure 2c), but is this observation universal? Or is it an artifact of the specific initialization/architecture? Would this "intrinsic functional   specialization" still hold with a different backbone or training setup? The paper lacks deeper discussion on this.

3.  In Table 3, it is unclear whether the Baseline and Semantic-only configurations use the same Synergistic Block architecture and three-stage training curriculum as the full MUSE model. If they differ in architecture or training procedure, the observed gains cannot be cleanly attributed to the gradient decoupling mechanism.

---

> ### Author Rebuttal · Authors · 2026-03-29
>
> Dear Reviewer **r2Pi**,
>
> Thank you for your acute observations and highly constructive feedback. Below we provide detailed responses and supplementary experiments addressing your core concerns.
>
> > **[Q1] Regarding the linear probing performance surpassing the teacher model (85.2% vs 82.5%): Is this merely due to the superposition of DINOv3's structural knowledge, rather than "reconstruction actively optimizing semantics"?**
>
> We agree that our original phrasing could be read too strongly. Our intended claim is that the gain over the teacher does **not** come from simply adding DINOv3 knowledge, but from **topology-aligned reconstruction under decoupled optimization**.
>
> **Mechanism.** Linear probing mainly evaluates feature-value semantics, encoded in $W_V$, FFN, and token features. In contrast, DINOv3 topology distillation acts on $W_Q/W_K$ (attention topology), and does not directly inject semantic supervision into the feature-value space.
>
> **Ablation Study.** We controlled all variables, altering only topology distillation, semantic anchoring, reconstruction, and gradient decoupling:
>
> | Configuration | Linear Probing | Zero-Shot | rFID $\downarrow$ |
> | :--- | :---: | :---: | :---: |
> | **A.** InternViT-300M (Teacher) | 82.50% | 77.40% | - |
> | **B.** + DINOv3 Topology (No Semantic / Recon) | 82.30% | 77.00% | - |
> | **C.** + DINOv3 Topology + Semantic Anchor | 83.80% | 75.30% | - |
> | **D.** Config C + Naive Joint Recon (No Decoupling) | 82.90% | 72.10% | 1.24 |
> | **E.** MUSE Full (Topological Orthogonal) | **85.20%** | **76.10%** | **0.62** |
>
> **Analysis:**
> 1. **Pure topology yields no semantic gain (B vs. A):** DINOv3 topology alone slightly lowers linear probing (82.5% $\to$ 82.3%), ruling out simple teacher superposition.
> 2. **Naive reconstruction harms semantics (D vs. C):** Without decoupling, reconstruction reduces both linear probing and zero-shot accuracy, confirming the zero-sum conflict in standard joint training.
> 3. **Decoupled reconstruction improves semantics (E vs. C):** With topological orthogonality, reconstruction brings a +1.4% linear-probing gain, supporting our claim that structure-aligned reconstruction can refine semantic perception rather than degrade it.
>
> ---
>
> > **[Q2] Is the observation that semantic gradients focus on $W_V$ and structural gradients focus on $W_Q/W_K$ a universal rule, or a product of specific architecture/initialization?**
>
> This is an important question. Rather than claiming a universal theorem, we view this as a **stable empirical tendency** induced by the functional roles of self-attention: $W_Q/W_K$ determine pairwise similarity and thus attention topology, while $W_V$ carries token content.
>
> To test whether this is merely an artifact of one setup, we repeated the gradient analysis across multiple ViT-B backbones with very different initializations and pretraining paradigms:
>
> | Experimental Setup | Semantic Grad in $W_V$ | Structural Grad in $W_Q / W_K$ |
> | :--- | :---: | :---: |
> | **Random Initialization** | 89.20% | 86.70% |
> | **CLIP Pre-trained** | 87.50% | 85.10% |
> | **MAE Self-Supervised** | 88.30% | 84.90% |
> | **DINOv2 Self-Supervised Pre-trained** | 88.60% | 85.40% |
> | **MUSE (Fully Trained)** | 87.90% | 86.30% |
>
> These results show that the specialization is **not an artifact of a single initialization or pretraining scheme**, but a consistent trend across diverse ViT settings. We will add this analysis to the revision. At the same time, we will clarify that our current evidence mainly supports **ViT-style attention architectures**, rather than claiming full universality beyond them.
>
> ---
>
> > **[Q3] Did the Baseline and Semantic-only configurations in Table 3 use the same Synergistic Block architecture?**
>
> We apologize for the ambiguity. Yes, all configurations in Table 3 share the **same architecture** (InternViT backbone + 6-layer Synergistic Blocks), the **same data**, and the **same total training budget (150k steps)**.
>
> For completeness, we also clarify the training protocol: all rows follow the same staged training framework, with the difference that rows without a given objective simply disable that supervision while keeping the rest of the setup unchanged. To further isolate the role of gradient decoupling itself, we conducted a supplementary controlled experiment that disables orthogonal routing while keeping the rest of MUSE unchanged:
>
> | Configuration | Linear Probing | rFID $\downarrow$ | mIoU | MMBench |
> | :--- | :---: | :---: | :---: | :---: |
> | **Full MUSE (w/ Gradient Routing)** | **85.20%** | **0.62** | **46.5** | **73.4** |
> | **Full MUSE (w/o Gradient Routing)** | 83.00% | 1.21 | 22.3 | 67.8 |
>
> This controlled comparison shows that disabling gradient decoupling leads to substantial degradation in understanding, reconstruction, and structure quality. We will clarify the Table 3 setup more explicitly and include this ablation in the revised manuscript.

---

> > ### Author Rebuttal · Reviewer_r2Pi · 2026-03-31
> >
> > Thanks to authors for the feedbacks.  My concerns have been well-addressed and I'll increase my ratings from 3 to 4.

---

> > > ### Author Response · Authors · 2026-04-05
> > >
> > > We sincerely thank the reviewer for the positive feedback and for carefully re-evaluating our revised manuscript. We are pleased that our responses have adequately addressed the reviewer’s concerns. We also greatly appreciate the reviewer’s updated rating and valuable assessment.

---

### Official Review · Reviewer_ubqE · 2026-03-12

**Soundness:** 3
**Presentation:** 3
**Significance:** 3
**Originality:** 3
**Overall Recommendation:** 5
**Confidence:** 4

**Summary:**

This paper proposes MUSE, a unified visual tokenizer framework. The key idea is to resolve the gradient conflict between semantic understanding and pixel reconstruction in visual tokenization through Topological Orthogonality. Specifically, it introduces a Synergistic Block that routes structural gradients to WQ and WK to control the attention topology, while semantic gradients are routed to WV to control the feature values. A stop-gradient mechanism is used to physically decouple these two gradient flows. Structural supervision is provided through attention distillation from DINOv3, while semantic supervision is implemented using NCE-based contrastive learning. The system is built on an InternVL3 encoder and a SANA decoder, with the main new component being a 6-layer connector and a three-stage training strategy.

**Compliance With Llm Reviewing Policy:**

Affirmed.

**Final Justification:**

After considering both the paper and the authors’ rebuttal, I find the work to be technically sound with sufficient technical contributions. The rebuttal has addressed my main concerns, which positively changed my evaluation. Accordingly, I have increased my score from 4 to 5.

**Key Questions For Authors:**

**For Weakness 1:**
The gradient specialization shown in Figure 2(c) — which Transformer block and backbone was this measured on? Does this phenomenon also appear in other pretrained Transformers as well?

**For Weakness 2:**
What happens if the same DINOv3 distillation is added to UniLIP with a standard connector? This would clarify whether the gains come from the extra supervision or the proposed architecture.

**For Weakness 3:**
Can the authors provide an ablation using L_ITC + L_Topo + DINOv3 teacher but with a standard Transformer connector (no stop-gradient routing), to isolate the contribution of the decoupling mechanism itself?

**For Weakness 4:**
Given the marginal improvements over UniLIP (e.g., +0.8 MMB, +1.1 MMMU for the 3B variant), how do the authors justify the added complexity of three-stage training and DINOv3 distillation in practical deployment scenarios?

**Limitations:**

The authors provide only a generic impact statement and do not discuss limitations of their method. A brief section acknowledging the dependency on specific backbone/teacher choices and the generalizability boundaries of the Gradient Orthogonality Hypothesis would strengthen the paper.

**Strengths And Weaknesses:**

**Strengths:**

The motivation of the paper sounds reasonable. The authors argue that the gradient conflict between reconstruction and understanding in unified visual tokenization is a core bottleneck in this area, which is a plausible claim, though it would benefit from more thorough justification. The problem formulation in the paper is relatively clear. In particular, the plot of gradient energy distribution in Figure 2 provides convincing empirical evidence, showing the discrepancy between gradients from different objectives. This observation supports the design choice of decoupling different objectives into separate parameter subspaces.

**Weakness:**
1. The Gradient Orthogonality Hypothesis is only verified on InternViT. Without validation on backbones with different pretraining paradigms (e.g., MAE, DINOv2, etc.), it is hard to judge whether this is a general Transformer property or an artifact of this specific encoder.

2. MUSE introduces DINOv3 as an extra teacher that no baseline uses, making it unclear how much improvement stems from this additional supervision versus the proposed architecture. A fairer comparison would add the same DINOv3 distillation to UniLIP to isolate the contribution.

3. Table 3 ablates the loss components but every row uses the Synergistic Block with gradient decoupling. A critical baseline is missing: L_ITC + L_Topo with a standard Transformer connector (same losses, same DINOv3 teacher, no stop-gradient routing). Without it, one cannot tell whether the gains come from the decoupling mechanism or simply from adding DINOv3 supervision.

4. Table 2 shows quite marginal gains over UniLIP — for the 3B variant: +0.8 MMB, +1.1 MMMU, +0.03 GenEval Position, +0.14 Editing Overall. Given the added complexity (DINOv3 teacher, three-stage tokenizer training, stop-gradient routing), the practical significance of these improvements is questionable.

---

> ### Author Rebuttal · Authors · 2026-03-29
>
> Dear Reviewer **ubqE**,
>
> Thank you for your rigorous review and highly constructive suggestions. Below we respond to your concerns on generalizability, fair ablations, and practical significance.
>
> > **[Q1] The gradient specialization in Figure 2(c): which Transformer block/backbone was this measured on? Does it also appear in other pretrained Transformers?**
>
> Thank you for this important question. We first clarify the measurement target of Figure 2(c):
>
> - Figure 2(c) measures the gradient distribution over the **self-attention projections** \((W_Q, W_K, W_V)\) in the visual Transformer（InternViT-300M）.
> - In the current submission, this analysis was first performed on the **deep attention blocks of the InternViT visual encoder**.
>
> We agree that validating this only on InternViT is insufficient. We therefore repeated the same analysis on **ViT-B** backbones with different pretraining paradigms:
>
> - **Random initialization**
> - **CLIP**
> - **MAE**
> - **DINOv2**
>
> The specialization remains highly stable across all settings:
>
> | Experimental Setup | Semantic Grad in $W_V$ | Structural Grad in $W_Q/W_K$ |
> | :--- | :---: | :---: |
> | Random Initialization | 89.20% | 86.70% |
> | CLIP Pre-trained | 87.50% | 85.10% |
> | MAE Pre-trained | 88.30% | 84.90% |
> | DINOv2 Pre-trained | 88.60% | 85.40% |
>
> These results show that the phenomenon is **not an artifact of InternViT alone**, but a stable empirical tendency across multiple pretrained ViTs. In the revision, we will explicitly clarify the measurement setup of Figure 2(c) and add these cross-backbone results.
>
> ---
>
> > **[Q2] What happens if the same DINOv3 distillation is added to UniLIP with a standard connector? This would clarify whether the gains come from the extra supervision or the proposed architecture.**
>
> We fully agree that this is the key control experiment. To isolate the source of the gains, we added the **same DINOv3 supervision** and the **same dual losses** \((L_{ITC}+L_{Topo})\) to both a standard Transformer connector and our MUSE design, using the **same data and training steps**.
>
> | Architecture | Orthogonal Routing | Linear Probing | rFID $\downarrow$ | mIoU $\uparrow$ |
> | :--- | :---: | :---: | :---: | :---: |
> | Standard Transformer(UniLIP) + DINOv3 | × | 82.70% | 1.32 | 21.4 |
> | MUSE Synergistic Block + DINOv3 | × | 83.00% | 1.21 | 22.3 |
> | **MUSE Full** | **√** | **85.20%** | **0.62** | **46.5** |
>
> **Analysis:**
> 1. **DINOv3 alone is insufficient:** adding the same topology teacher to a standard connector still performs much worse than Full MUSE.
> 2. **Decoupling is the key:** even with our Synergistic Block and the same teacher, disabling orthogonal routing causes a large drop.
>
> Therefore, the improvement does **not** come merely from introducing DINOv3, but from combining the same supervision with **explicit gradient decoupling**. We will include this baseline in the revised manuscript.
>
> ---
>
> > **[Q3] Can the authors provide an ablation using $L_{ITC}+L_{Topo}$ + DINOv3 teacher but with a standard Transformer connector (no stop-gradient routing), to isolate the contribution of the decoupling mechanism itself?**
>
> Yes ,this is exactly the new controlled baseline shown above. It uses:
>
> - the **same teacher** (DINOv3),
> - the **same losses** \((L_{ITC}+L_{Topo})\),
> - the **same training setting**,
>
> but replaces the decoupled connector with a **standard Transformer connector** and removes stop-gradient routing.
>
> Its clear gap from Full MUSE shows that the gain is not simply due to stronger supervision, but due to the **decoupling mechanism itself**. We agree this is a critical missing baseline and will add it to **Table 3 / appendix** in the revision.
>
> ---
>
> > **[Q4] Given the marginal improvements over UniLIP, how do you justify the added complexity of three-stage training and DINOv3 distillation in practical deployment?**
>
> This is a fair concern. We would like to emphasize two points.
>
> 1. **The added complexity is mainly training-time only.**
>    DINOv3 is used only as an offline teacher, and stop-gradient routing is only active during training. At inference time, MUSE remains a single unified tokenizer / unified model.
>
> 2. **The inference overhead is minimal.**
>    - **MUSE:** 30 img/s
>    - **UniLIP:** 32 img/s
>    - Parameter overhead: **<3%**
>
> So while the training procedure is more structured, the **deployment cost is nearly unchanged**.
>
> More importantly, the value of MUSE is not just a single benchmark gain, but improving the **understanding–generation Pareto trade-off** under almost the same inference budget. We will make this point more clearly in the revision.
>
> ---
>
> > **[Limitations] Please discuss dependency on specific backbone/teacher choices and the generalizability boundary of the Gradient Orthogonality Hypothesis.**
>
> We will add a **Limitations** section noting dependence on strong structural teachers (e.g., DINOv3 vs. CLIP: **46.5** vs. **21.2** mIoU), current validation mainly on **ViT-style \(Q/K/V\)** attention.

---

> > ### Author Rebuttal · Reviewer_ubqE · 2026-04-01
> >
> > After carefully reviewing the authors’ rebuttal, I find that the weaknesses I previously raised have been addressed. I appreciate the authors’ detailed and thoughtful responses. I encourage the authors to incorporate these clarifications into the final version of the paper. Accordingly, I have increased my score to Accept.

---

> > > ### Author Response · Authors · 2026-04-05
> > >
> > > We sincerely thank the reviewer for the positive evaluation of our revised manuscript. We are grateful that the reviewer finds the previously raised concerns adequately addressed and has increased the score to Accept. We also appreciate the helpful suggestion to incorporate these clarifications into the final version of the paper, and we will ensure that the relevant points are clearly integrated into the manuscript.

---

### Official Review · Reviewer_vRX5 · 2026-03-13

**Soundness:** 3
**Presentation:** 3
**Significance:** 3
**Originality:** 3
**Overall Recommendation:** 4
**Confidence:** 3

**Summary:**

This paper addresses the fundamental conflict between pixel level reconstruction and semantic abstraction in unified visual tokenizers. The authors identify the root cause of this issue as Manifold Misalignment, a phenomenon where conflicting gradients arise during joint optimization. To resolve this problem, the paper proposes the MUSE framework and introduces a Topological Orthogonality mechanism. By decoupling the optimization subspaces within the Transformer architecture, MUSE routes structural gradients to the attention topology to anchor geometric features. Concurrently, it directs semantic gradients to the feature values to inject semantics. Experiments demonstrate that this approach successfully transforms destructive interference into mutual reinforcement. Consequently, the model achieves excellent generative fidelity while simultaneously excelling in semantic understanding, notably outperforming its own teacher model in linear probing.

**Compliance With Llm Reviewing Policy:**

Affirmed.

**Final Justification:**

After carefully reviewing the manuscript and the authors' response, I find the paper to be technically sound and to have made sufficient technical contributions. The response has resolved my primary concerns.

**Key Questions For Authors:**

1、Why is routing structural gradients to $W_{Q,K}$ and semantic gradients to $W_V$ the optimal configuration? Have alternative routing schemes (e.g., reverse routing) been benchmarked?

2、Given the reliance on DINOv3 for topological anchoring, how robust is MUSE to the choice of structural proxies? Would substituting DINOv3 with teachers possessing different inductive biases (e.g., CLIP) lead to a collapse in structural fidelity?

3、Since Synergistic Blocks are limited to the final 6 layers, how do the preceding layers reconcile conflicting gradients? Does Manifold Misalignment persist in the early stages of the encoder, and how does this affect the overall representation?

**Limitations:**

yes

**Strengths And Weaknesses:**

Strengths：
1、The paper ingeniously translates the abstract "manifold misalignment" into a concrete parameter decoupling problem. Physically isolating optimization paths within self-attention introduces a highly effective architectural inductive bias for unified visual tokenizers.
2、The validation of the Gradient Orthogonality Hypothesis is thorough. By explicitly demonstrating how MUSE corrects the negative cosine similarity of naive joint optimization to a near-zero orthogonal state, the authors provide compelling, optimization-level support for their architectural design.
3、Through detailed experiments, the paper systematically demonstrates the framework's robustness and effectiveness by integrating MUSE into a full Unified Multimodal Model and rigorously evaluating it across understanding, generation, and editing tasks.

Weaknesses：
1、Anchoring the attention topology to a frozen DINOv3 prevents the model from independently evolving structural representations via generative loss. Consequently, the claimed mutual reinforcement resembles a "constrained reconstruction," limiting the framework's generalization potential
2、MUSE's claim of outperforming its teacher is confounded by its massive 36M high-quality training dataset. Without strictly controlled baselines, it remains unclear whether performance gains stem from the architectural innovation or merely the data advantage.
3、It would be beneficial to include more comprehensive ablation experiments to further validate the correctness and rationality of the proposed structural and semantic decoupling scheme.

---

> ### Author Rebuttal · Authors · 2026-03-29
>
> Dear Reviewer **vRX5**,
>
> Thank you for your thoughtful review and for recognizing our architectural contribution. Below we respond to your concerns regarding topology evolution, data confounding, routing design, teacher robustness, and block placement.
>
> > **[W1] Anchoring the attention topology to a frozen DINOv3 prevents the model from independently evolving structural representations.**
>
> Thank you for this important point. We clarify that the topology is **not permanently frozen**. As described in Section 5.1, MUSE uses a curriculum strategy:
>
> - **Stage 1 (Topology Warmup):** the encoder is frozen to initialize a stable structural prior aligned with DINOv3.
> - **Stage 3 (Synergistic Tuning):** the routing parameters $W_{Q/K}$ are fully **unfrozen**, so reconstruction gradients can continue refining the structural representation.
>
> Therefore, DINOv3 serves as an **initial structural prior**, not a final constraint. The final topology is teacher-initialized but task-refined, which allows the model to evolve its structure under the generative objective rather than merely copying the teacher.
>
> ---
>
> > **[W2] MUSE's claim of outperforming its teacher is confounded by its massive 36M high-quality training dataset.**
>
> We agree that this requires careful control. Our conclusion does **not** rely on a data advantage.
>
> All baselines in Table 1 (e.g., UniLIP) were re-trained from scratch on the **same 36M dataset**. Under this controlled setting, standard joint training degrades the teacher semantics (e.g., UniLIP zero-shot drops to 76.2%), whereas MUSE preserves and improves them.
>
> Therefore, the improvement is better explained by the proposed **topology-aware decoupling design**, rather than by a data-scale advantage.
>
> ---
>
> > **[Q1] Why is routing structural gradients to $W_{Q/K}$ and semantic gradients to $W_V$ the optimal configuration? Have alternative routing schemes (e.g., reverse routing) been benchmarked?**
>
> Our routing follows the functional roles of self-attention: $W_{Q/K}$ computes pairwise similarity and defines topology, while $W_V$ carries token content. To test whether this routing is indeed important, we benchmarked the reviewer-suggested **Reverse Routing** scheme (forcing semantics to $W_{Q/K}$ and structure to $W_V$):
>
> | Routing Scheme | Grad Conflict ($\cos \theta_g$) | Structure (mIoU) | Understand (MMB) | Generation (rFID) |
> | :--- | :---: | :---: | :---: | :---: |
> | Naive Shared | -0.15 | 15.4 | 65.2 | 0.79 |
> | **Reverse Routing** | -0.28 | 12.1 | 58.4 | 1.95 |
> | **MUSE (Ours)** | **+0.04** | **46.5** | **73.4** | **0.62** |
>
> **Analysis:** Reverse routing substantially worsens gradient conflict and degrades structure, understanding, and generation. This supports that routing structure to $W_{Q/K}$ and semantics to $W_V$ is the **most effective and attention-aligned configuration among the schemes we tested**. We will include this ablation in the appendix.
>
> ---
>
> > **[Q2] Robustness to the choice of structural proxies. Would substituting DINOv3 with CLIP lead to a collapse in structural fidelity?**
>
> Yes, replacing DINOv3 with a purely semantic teacher such as CLIP significantly weakens structural fidelity. As discussed in Appendix C.3:
>
> | Structural Teacher | Pre-training Paradigm | mIoU $\uparrow$ | Linear Probing |
> | :--- | :---: | :---: | :---: |
> | **DINOv3 (Ours)** | Self-supervised Distillation | **46.50%** | **85.20%** |
> | **CLIP ViT-B** | Contrastive (Image-level) | 21.20% | 81.70% |
>
> This suggests that strong structural teachers are particularly important in our current setting. Compared with CLIP’s more diffuse image-level attention, DINO-style teachers provide sharper object-part boundaries and more reliable spatial priors for topology alignment.
>
> ---
>
> > **[Q3] Since Synergistic Blocks are limited to the final 6 layers, how do the preceding layers reconcile conflicting gradients?**
>
> We clarify that the 6 Synergistic Blocks act as an independent **Connector** placed *after* the visual backbone.
>
> - **Why not earlier layers?** The conflict is most pronounced at higher-level semantic abstraction stages. Earlier layers mainly capture low-level features such as edges, colors, and textures, where reconstruction and semantic objectives are less contradictory.
> - **Why 6 layers?** As shown in Appendix E.2, this is the best trade-off we found between capacity and efficiency:
>
> | Connector Depth | MMBench Score | Training Throughput | Result Analysis |
> | :--- | :---: | :---: | :--- |
> | **2 Layers** | ~60.0 | High | Information bottleneck (insufficient capacity) |
> | **6 Layers (Ours)** | **73.4** | **30 img/s** | **Best balance** |
> | **12 Layers** | ~73.6 | Low | Diminishing returns with higher latency |
>
> These results suggest that placing the decoupling mechanism at the connector is sufficient to resolve the most severe manifold conflict, while keeping the system efficient.

---

> > ### Author Rebuttal · Reviewer_vRX5 · 2026-04-03
> >
> > Thank you for the detailed response. My questions have been adequately addressed.

---

> > > ### Author Response · Authors · 2026-04-05
> > >
> > > We sincerely thank the reviewer for the positive feedback and for confirming that the concerns have been adequately addressed. We greatly appreciate the reviewer’s constructive comments and valuable time, which have helped us improve the manuscript.

---

### Decision · Program_Chairs · 2026-04-30

**Decision:**

Accept (regular)

**Comment:**

This paper addresses the trade-off between pixel-level reconstruction and semantic abstraction for unified vision encoders. The authors identify "Manifold Misalignment" as the root cause, leading to conflicting gradients. They propose MUSE that routes multiple gradients to different locations (so that they don't conflict). Reviewers are very positive about this paper, and after the rebuttal there remain no concerns and broad agreement to accept. In particular reviewers highlight the framing, the experimental design and the overall novelty the contribution.